# A Deep Recurrent Neural Network via Unfolding Reweighted $\ell_1$-$\ell_1$ Minimization

## Abstract

Deep unfolding methods design deep neural networks as learned variations of optimization methods. These networks have been shown to achieve faster convergence and higher accuracy than the original optimization methods. In this line of research, this paper develops a novel deep recurrent neural network (coined reweighted-RNN) by unfolding a reweighted $\ell_1$-$\ell_1$ minimization algorithm and applies it to the task of sequential signal reconstruction. To the best of our knowledge, this is the first deep unfolding method that explores reweighted minimization. Due to the underlying reweighted minimization model, our RNN has a different soft-thresholding function (alias, different activation function) for each hidden unit in each layer. Furthermore, it has higher network expressivity than existing deep unfolding RNN models due to the over-parameterizing weights. Moreover, we establish theoretical generalization error bounds for the proposed reweighted-RNN model by means of Rademacher complexity. The bounds reveal that the parameterization of the proposed reweighted-RNN ensures good generalization. We apply the proposed reweighted-RNN to the problem of video-frame reconstruction from low-dimensional measurements, that is, sequential frame reconstruction. The experimental results on the moving MNIST dataset demonstrate that the proposed deep reweighted-RNN significantly outperforms existing RNN models.

## 1 Introduction

The problem of reconstructing sequential signals from low-dimensional measurements across time is of great importance for a number of applications such as time-series data analysis, future-frame prediction, and compressive video sensing. Specifically, we consider the problem of reconstructing a sequence of signals $\mathbf{s}_t \in \mathbb{R}^{n_0}$, $t = 1, 2, \dots, T$, from low-dimensional measurements $\mathbf{x}_t = \mathbf{A}\mathbf{s}_t$, where $\mathbf{A} \in \mathbb{R}^{n \times n_0}$ ($n \ll n_0$) is a sensing matrix. We assume that $\mathbf{s}_t$ has a sparse representation $\mathbf{h}_t \in \mathbb{R}^h$ in a dictionary $\mathbf{D} \in \mathbb{R}^{n_0 \times h}$, that is, $\mathbf{s}_t = \mathbf{D}\mathbf{h}_t$. At each time step $t$, the signal $\mathbf{s}_t$ can be independently reconstructed using the measurements $\mathbf{x}_t$ by solving (Donoho, 2006):

$$\min_{\mathbf{h}_t} \left\{ \frac{1}{2} \|\mathbf{x}_t - \mathbf{A}\mathbf{D}\mathbf{h}_t\|_2^2 + \lambda \|\mathbf{h}_t\|_1 \right\}, \tag{1}$$

where $\|\cdot\|_p$ is the $\ell_p$-norm and $\lambda$ is a regularization parameter. The iterative shrinkage-thresholding algorithm (ISTA) (Daubechies et al., 2004) solves (1) by iterating over $\mathbf{h}_t^{(l)} = \phi_{\frac{\lambda}{c}}(\mathbf{h}_t^{(l-1)} - \frac{1}{c}\mathbf{D}^{\mathrm{T}}\mathbf{A}^{\mathrm{T}}(\mathbf{A}\mathbf{D}\mathbf{h}_t^{(l-1)} - \mathbf{x}_t))$, where $l$ is the iteration counter, $\phi_\gamma(u) = \text{sign}(u)[0, |u| - \gamma]_+$ is the soft-thresholding operator, $\gamma = \frac{\lambda}{c}$, and $c$ is an upper bound on the Lipschitz constant of the gradient of $\frac{1}{2}\|\mathbf{x}_t - \mathbf{A}\mathbf{D}\mathbf{h}_t\|_2^2$.

Under the assumption that sequential signal instances are correlated, we consider the following sequential signal reconstruction problem:

$$\min_{\mathbf{h}_t} \left\{ \frac{1}{2} \|\mathbf{x}_t - \mathbf{A}\mathbf{D}\mathbf{h}_t\|_2^2 + \lambda_1 \|\mathbf{h}_t\|_1 + \lambda_2 R(\mathbf{h}_t, \mathbf{h}_{t-1}) \right\}, \tag{2}$$

where $\lambda_1, \lambda_2 > 0$ are regularization parameters and $R(\mathbf{h}_t, \mathbf{h}_{t-1})$ is an added regularization term that expresses the similarity of the representations $\mathbf{h}_t$ and $\mathbf{h}_{t-1}$ of two consecutive signals. Wisdom et al. (2017) proposed an RNN design (coined Sista-RNN) by unfolding the sequential version of

ISTA. That study assumed that two consecutive signals are close in the $\ell_2$-norm sense, formally, $R(\mathbf{h}_t, \mathbf{h}_{t-1}) = \frac{1}{2}\|\mathbf{D}\mathbf{h}_t - \mathbf{F}\mathbf{D}\mathbf{h}_{t-1}\|_2^2$, where $\mathbf{F} \in \mathbb{R}^{n_0 \times n_0}$ is a correlation matrix between $\mathbf{s}_t$ and $\mathbf{s}_{t-1}$. More recently, the study by Le et al. (2019) designed the $\ell_1$-$\ell_1$-RNN, which stems from unfolding an algorithm that solves the $\ell_1$-$\ell_1$ minimization problem (Mota et al., 2017; 2016). This is a version of Problem (2) with $R(\mathbf{h}_t, \mathbf{h}_{t-1}) = \|\mathbf{h}_t - \mathbf{G}\mathbf{h}_{t-1}\|_1$, where $\mathbf{G} \in \mathbb{R}^{h \times h}$ is an affine transformation that promotes the correlation between $\mathbf{h}_t$ and $\mathbf{h}_{t-1}$. Both studies (Wisdom et al., 2017; Le et al., 2019) have shown that carefully-designed deep RNN models outperform the generic RNN model and ISTA (Daubechies et al., 2004) in the task of sequential frame reconstruction.

Deep neural networks (DNN) have achieved state-of-the-art performance in solving (1) for individual signals, both in terms of accuracy and inference speed (Mousavi et al., 2015). However, these models are often criticized for their lack of interpretability and theoretical guarantees (Lucas et al., 2018). Motivated by this, several studies focus on designing DNNs that incorporate domain knowledge, namely, signal priors. These include deep unfolding methods which design neural networks to learn approximations of iterative optimization algorithms. Examples of this approach are LISTA (Gregor & LeCun, 2010) and its variants, including ADMM-Net (Sun et al., 2016), AMP (Borgerding et al., 2017), and an unfolded version of the iterative hard thresholding algorithm (Xin et al., 2016).

LISTA (Gregor & LeCun, 2010) unrolls the iterations of ISTA into a feed-forward neural network with weights, where each layer implements an iteration: $\mathbf{h}_t^{(l)} = \phi_{\gamma^{(l)}}(\mathbf{W}^{(l)}\mathbf{h}_t^{(l-1)} + \mathbf{U}^{(l)}\mathbf{x}_t)$, with $\mathbf{W}^{(l)} = \mathbf{I} - \frac{1}{c}\mathbf{D}^{\mathrm{T}}\mathbf{A}^{\mathrm{T}}\mathbf{A}\mathbf{D}$, $\mathbf{U}^{(l)} = \frac{1}{c}\mathbf{D}^{\mathrm{T}}\mathbf{A}^{\mathrm{T}}$, and $\gamma^{(l)}$ being learned from data. It has been shown (Gregor & LeCun, 2010; Sprechmann et al., 2015) that a $d$-layer LISTA network with trainable parameters $\boldsymbol{\Theta} = \{\mathbf{W}^{(l)}, \mathbf{U}^{(l)}, \gamma^{(l)}\}_{l=1}^d$ achieves the same performance as the original ISTA but with much fewer iterations (i.e., number of layers). Recent studies (Chen et al., 2018; Liu et al., 2019) have found that exploiting dependencies between $\mathbf{W}^{(l)}$ and $\mathbf{U}^{(l)}$ leads to reducing the number of trainable parameters while retaining the performance of LISTA. These works provided theoretical insights to the convergence conditions of LISTA. However, the problem of *designing deep unfolding methods for dealing with sequential signals* is significantly less explored. In this work, we will consider a deep RNN for solving Problem (2) that outputs a sequence, $\hat{\mathbf{s}}_1, \ldots, \hat{\mathbf{s}}_T$ from an input measurement sequence, $\mathbf{x}_1, \ldots, \mathbf{x}_T$, as following:

$$\begin{aligned} \mathbf{h}_t &= \phi_\gamma(\mathbf{W}\mathbf{h}_{t-1} + \mathbf{U}\mathbf{x}_t), \\ \hat{\mathbf{s}}_t &= \mathbf{D}\mathbf{h}_t. \end{aligned} \tag{3}$$

It has been shown that reweighted algorithms—such as the reweighted $\ell_1$ minimization method by Candès et al. (2008) and the reweighted $\ell_1$-$\ell_1$ minimization by Luong et al. (2018)—outperform their non-reweighted counterparts. Driven by this observation, this paper proposes a novel deep RNN architecture by unfolding a reweighted-$\ell_1$-$\ell_1$ minimization algorithm. Due to the reweighting, our network has higher expressivity than existing RNN models leading to better data representations, especially when depth increases. This is in line with recent studies (He et al., 2016; Cortes et al.; Huang et al., 2017), which have shown that better performance can be achieved by highly over-parameterized networks, i.e., networks with far more parameters than the number of training samples. While the most recent studies (related over-parameterized DNNs) consider fully-connected networks applied on classification problems (Neyshabur et al., 2019), our approach focuses on deep-unfolding architectures and opts to understand how the networks learn a low-complexity representation for sequential signal reconstruction, which is a regression problem across time. Furthermore, while there have been efforts to build deep RNNs (Pascanu et al., 2014; Li et al., 2018; Luo et al., 2017; Wisdom et al., 2017), examining the generalization property of such deep RNN models on unseen sequential data still remains elusive. In this work, we derive the generalization error bound of the proposed design and further compare it with existing RNN bounds (Zhang et al., 2018; Kusupati et al., 2018).

**Contributions**. The contributions of this work are as follows:

- We propose a principled deep RNN model for sequential signal reconstruction by unfolding a reweighted $\ell_1$-$\ell_1$ minimization method. Our reweighted-RNN model employs different soft-thresholding functions that are adaptively learned per hidden unit. Furthermore, the proposed model is over-parameterized, has high expressivity and can be efficiently stacked.

- We derive the generalization error bound of the proposed model (and deep RNNs) by measuring Rademacher complexity and show that the over-parameterization of our RNN ensures good generalization. To best of our knowledge, this is the first generalization error

bound for deep RNNs; moreover, our bound is tighter than existing bounds derived for shallow RNNs (Zhang et al., 2018; Kusupati et al., 2018).

- We provide experiments in the task of reconstructing video sequences from low-dimensional measurements. We show significant gains when using our model compared to several state-of-the-art RNNs (including unfolding architectures), especially when the depth of RNNs increases.

## 2 A DEEP RNN VIA UNFOLDING REWEIGHTED-$\ell_1$-$\ell_1$ MINIMIZATION

In this section, we describe a reweighted $\ell_1$-$\ell_1$ minimization problem for sequential signal reconstruction and propose an iterative algorithm based on the proximal method. We then design a deep RNN architecture by unfolding this algorithm.

**The proposed reweighted $\ell_1$-$\ell_1$ minimization**. We introduce the following problem:

$$\min_{\mathbf{h}_t} \left\{ \frac{1}{2}\|\mathbf{x}_t - \mathbf{ADZh}_t\|_2^2 + \lambda_1\|\mathbf{g} \circ \mathbf{Zh}_t\|_1 + \lambda_2\|\mathbf{g} \circ (\mathbf{Zh}_t - \mathbf{Gh}_{t-1})\|_1 \right\}, \tag{4}$$

where "∘" denotes element-wise multiplication, $\mathbf{g} \in \mathbb{R}^h$ is a vector of positive weights, $\mathbf{Z} \in \mathbb{R}^{h \times h}$ is a reweighting matrix, and $\mathbf{G} \in \mathbb{R}^{h \times h}$ is an affine transformation that promotes the correlation between $\mathbf{h}_{t-1}$ and $\mathbf{h}_t$. Intuitively, $\mathbf{Z}$ is adopted to transform $\mathbf{h}_t$ to $\mathbf{Zh}_t \in \mathbb{R}^h$, producing a reweighted version of it. Thereafter, $\mathbf{g}$ aims to reweight each transformed component of $\mathbf{Zh}_t$ and $\mathbf{Zh}_t - \mathbf{Gh}_{t-1}$ in the $\ell_1$-norm regularization terms. Because of applying reweighting (Candès et al., 2008), the solution of Problem (4) is a more accurate sparse representation compared to the solution of the $\ell_1$-$\ell_1$ minimization problem in Le et al. (2019) (where $\mathbf{Z} = \mathbf{I}$ and $\mathbf{g} = \mathbf{I}$). Furthermore, the use of the reweighting matrix $\mathbf{Z}$ to transform $\mathbf{h}_t$ to $\mathbf{Zh}_t$ differentiates Problem (4) from the reweighted $\ell_1$-$\ell_1$ minimization problem in Luong et al. (2018) where $\mathbf{Z} = \mathbf{I}$.

The objective function in (4) consists of the differentiable fidelity term $f(\mathbf{Zh}_t) = \frac{1}{2}\|\mathbf{x}_t - \mathbf{ADZh}_t\|_2^2$ and the non-smooth term $g(\mathbf{Zh}_t) = \lambda_1\|\mathbf{g} \circ \mathbf{Zh}_t\|_1 + \lambda_2\|\mathbf{g} \circ (\mathbf{Zh}_t - \mathbf{Gh}_{t-1})\|_1$. We use a proximal gradient method (Beck & Teboulle, 2009) to solve (4): At iteration $l$, we first update $\mathbf{h}_t^{(l-1)}$—after being multiplied by $\mathbf{Z}_l$—with a gradient descent step on the fidelity term as $\mathbf{u} = \mathbf{Z}_l\mathbf{h}_t^{(l-1)} - \frac{1}{c}\mathbf{Z}_l\nabla f(\mathbf{h}_t^{(l-1)})$, where $\nabla f(\mathbf{h}_t^{(l-1)}) = \mathbf{D}^T\mathbf{A}^T(\mathbf{ADh}_t^{(l-1)} - \mathbf{x}_t)$. Then, $\mathbf{h}_t^{(l)}$ is updated as

$$\mathbf{h}_t^{(l)} = \Phi_{\frac{\lambda_1}{c}\mathbf{g}_l, \frac{\lambda_2}{c}\mathbf{g}_l, \mathbf{Gh}_{t-1}}\left(\mathbf{Z}_l\mathbf{h}_t^{(l-1)} - \frac{1}{c}\mathbf{Z}_l\nabla f(\mathbf{h}_t^{(l-1)})\right), \tag{5}$$

where the proximal operator $\Phi_{\frac{\lambda_1}{c}\mathbf{g}_l, \frac{\lambda_2}{c}\mathbf{g}_l, \mathbf{Gh}_{t-1}}(\mathbf{u})$ is defined as

$$\Phi_{\frac{\lambda_1}{c}\mathbf{g}_l, \frac{\lambda_2}{c}\mathbf{g}_l, \hbar}(\mathbf{u}) = \arg\min_{\mathbf{v} \in \mathbb{R}^h}\left\{\frac{1}{c}g(\mathbf{v}) + \frac{1}{2}\|\mathbf{v} - \mathbf{u}\|_2^2\right\}, \tag{6}$$

with $\hbar = \mathbf{Gh}_{t-1}$. Since the minimization problem is separable, we can minimize (6) independently for each of the elements $g_l, \hbar, u$ of the corresponding $\mathbf{g}_l, \hbar, \mathbf{u}$ vectors. After solving (6), we obtain $\Phi_{\frac{\lambda_1}{c}g_l, \frac{\lambda_2}{c}g_l, \hbar}(u)$ [for solving (6), we refer to Proposition B.1 in Appendix B]. For $\hbar \geq 0$:

$$\Phi_{\frac{\lambda_1}{c}g_l, \frac{\lambda_2}{c}g_l, \hbar \geq 0}(u) = \begin{cases} u - \frac{\lambda_1}{c}g_l - \frac{\lambda_2}{c}g_l, & \hbar + \frac{\lambda_1}{c}g_l + \frac{\lambda_2}{c}g_l < u < \infty \\ \hbar, & \hbar + \frac{\lambda_1}{c}g_l - \frac{\lambda_2}{c}g_l \leq u \leq \hbar + \frac{\lambda_1}{c}g_l + \frac{\lambda_2}{c}g_l \\ u - \frac{\lambda_1}{c}g_l + \frac{\lambda_2}{c}g_l, & \frac{\lambda_1}{c}g_l - \frac{\lambda_2}{c}g_l < u < \hbar + \frac{\lambda_1}{c}g_l - \frac{\lambda_2}{c}g_l \\ 0, & -\frac{\lambda_1}{c}g_l - \frac{\lambda_2}{c}g_l \leq u \leq \frac{\lambda_1}{c}g_l - \frac{\lambda_2}{c}g_l \\ u + \frac{\lambda_1}{c}g_l + \frac{\lambda_2}{c}g_l, & -\infty < u < -\frac{\lambda_1}{c}g_l - \frac{\lambda_2}{c}g_l, \end{cases} \tag{7}$$

and for $\hbar < 0$:

$$\Phi_{\frac{\lambda_1}{c}g_l, \frac{\lambda_2}{c}g_l, \hbar < 0}(u) = \begin{cases} u - \frac{\lambda_1}{c}g_l - \frac{\lambda_2}{c}g_l, & \frac{\lambda_1}{c}g_l + \frac{\lambda_2}{c}g_l < u < \infty \\ 0, & -\frac{\lambda_1}{c}g_l + \frac{\lambda_2}{c}g_l \leq u \leq \frac{\lambda_1}{c}g_l + \frac{\lambda_2}{c}g_l \\ u + \frac{\lambda_1}{c}g_l - \frac{\lambda_2}{c}g_l, & \hbar - \frac{\lambda_1}{c}g_l + \frac{\lambda_2}{c}g_l < u < -\frac{\lambda_1}{c}g_l + \frac{\lambda_2}{c}g_l \\ \hbar, & \hbar - \frac{\lambda_1}{c}g_l - \frac{\lambda_2}{c}g_l \leq u \leq \hbar - \frac{\lambda_1}{c}g_l + \frac{\lambda_2}{c}g_l \\ u - \frac{\lambda_1}{c}g_l + \frac{\lambda_2}{c}g_l, & -\infty < u < \hbar - \frac{\lambda_1}{c}g_l - \frac{\lambda_2}{c}g_l \end{cases} \tag{8}$$

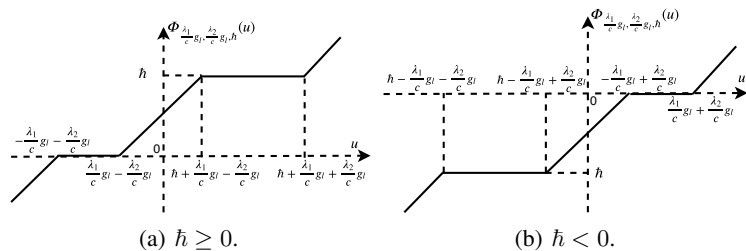

(a) $\hbar \geq 0$.    (b) $\hbar < 0$.

Figure 1: The generic form of the proximal operators for Algorithm 1 - but also the activation function in the proposed reweighted-RNN. Note that per unit per layer $g_l$ leads to a different activation function.

---

**Algorithm 1:** The proposed algorithm for sequential signal reconstruction.

1 **Input:** Measurements $\mathbf{x}_1, \ldots, \mathbf{x}_T$, measurement matrix $\mathbf{A}$, dictionary $\mathbf{D}$, affine transform $\mathbf{G}$,
   initial $\mathbf{h}_0^{(d)} \equiv \mathbf{h}_0$, reweighting matrices $\mathbf{Z}_1, \ldots, \mathbf{Z}_d$ and vectors $\mathbf{g}_1, \ldots, \mathbf{g}_d$, $c$, $\lambda_1$, $\lambda_2$.
2 **Output:** Sequence of sparse codes $\mathbf{h}_1, \ldots, \mathbf{h}_T$.
3 **for** $t = 1, \ldots, T$ **do**
4     $\mathbf{h}_t^{(0)} = \mathbf{G} \mathbf{h}_{t-1}^{(d)}$
5     **for** $l = 1$ to $d$ **do**
6        $\mathbf{u} = [\mathbf{Z}_l - \frac{1}{c}\mathbf{Z}_l\mathbf{D}^T\mathbf{A}^T\mathbf{A}\mathbf{D}]\mathbf{h}_t^{(l-1)} + \frac{1}{c}\mathbf{Z}_l\mathbf{D}^T\mathbf{A}^T\mathbf{x}_t$
7        $\mathbf{h}_t^{(l)} = \Phi_{\frac{\lambda_1}{c}\mathbf{g}_l, \frac{\lambda_2}{c}\mathbf{g}_l, \mathbf{G}\mathbf{h}_{t-1}^{(d)}}(\mathbf{u})$
8     **end**
9 **end**
10 **return** $\mathbf{h}_1^{(d)}, \ldots, \mathbf{h}_T^{(d)}$

---

Fig. 1 depicts the proximal operators for $\hbar \geq 0$ and $\hbar < 0$. Observe that different values of $g_l$ lead to different shapes of the proximal functions $\Phi_{\frac{\lambda_1}{c}\mathbf{g}_l, \frac{\lambda_2}{c}\mathbf{g}_l, \hbar}(u)$ for each element $u$ of $\mathbf{u}$.

Our iterative algorithm is given in Algorithm 1. We reconstruct a sequence $\mathbf{h}_1, \ldots, \mathbf{h}_T$ from a sequence of measurements $\mathbf{x}_1, \ldots, \mathbf{x}_T$. For each time step $t$, Step 6 applies a gradient descent update for $f(\mathbf{Z}\mathbf{h}_{t-1})$ and Step 7 applies the proximal operator $\Phi_{\frac{\lambda_1}{c}\mathbf{g}_l, \frac{\lambda_2}{c}\mathbf{g}_l, \mathbf{G}\mathbf{h}_{t-1}^{(d)}}$ element-wise to the result.

Let us compare the proposed method against the algorithm in Le et al. (2019)—which resulted in the $\ell_1$-$\ell_1$-RNN—that solves the $\ell_1$-$\ell_1$ minimization in Mota et al. (2016) (where $\mathbf{Z}_l = \mathbf{I}$ and $\mathbf{g}_l = \mathbf{I}$). In that algorithm, the update terms in Step 6, namely $\mathbf{I} - \frac{1}{c}\mathbf{D}^T\mathbf{A}^T\mathbf{A}\mathbf{D}$ and $\frac{1}{c}\mathbf{D}^T\mathbf{A}^T$, and the proximal operator in Step 7 are the same for all iterations of $l$. In contrast, Algorithm 1 uses a different $\mathbf{Z}_l$ matrix per iteration to reparameterize the update terms (Step 6) and, through updating $\mathbf{g}_l$, it applies a different proximal operator to each element $\mathbf{u}$ (in Step 7) per iteration $l$.

**The proposed reweighted-RNN.** We now describe the proposed architecture for sequential signal recovery, designed by unrolling the steps of Algorithm 1 across the iterations $l = 1, \ldots, d$ (yielding the hidden layers) and time steps $t = 1, \ldots, T$. Specifically, the $l$-th hidden layer is given by

$$\mathbf{h}_t^{(l)} = \begin{cases} \Phi_{\frac{\lambda_1}{c}\mathbf{g}_1, \frac{\lambda_2}{c}\mathbf{g}_1, \mathbf{G}\mathbf{h}_{t-1}^{(d)}}\left(\mathbf{W}_1\mathbf{h}_{t-1}^{(d)} + \mathbf{U}_1\mathbf{x}_t\right), & \text{if } l = 1, \\ \Phi_{\frac{\lambda_1}{c}\mathbf{g}_l, \frac{\lambda_2}{c}\mathbf{g}_l, \mathbf{G}\mathbf{h}_{t-1}^{(d)}}\left(\mathbf{W}_l\mathbf{h}_t^{(l-1)} + \mathbf{U}_l\mathbf{x}_t\right), & \text{if } l > 1, \end{cases} \tag{9}$$

and the reconstructed signal at time step $t$ is given by $\hat{\mathbf{s}}_t = \mathbf{D}\mathbf{h}_t^{(d)}$; where $\mathbf{U}_l$, $\mathbf{W}_l$, $\mathbf{V}$ are defined as

$$\mathbf{U}_l = \frac{1}{c}\mathbf{Z}_l\mathbf{D}^T\mathbf{A}^T, \forall l, \tag{10}$$

$$\mathbf{W}_1 = \mathbf{Z}_1\mathbf{G} - \frac{1}{c}\mathbf{Z}_1\mathbf{D}^T\mathbf{A}^T\mathbf{A}\mathbf{D}\mathbf{G}, \tag{11}$$

$$\mathbf{W}_l = \mathbf{Z}_l - \frac{1}{c}\mathbf{Z}_l\mathbf{D}^T\mathbf{A}^T\mathbf{A}\mathbf{D}, \ l > 1. \tag{12}$$

The activation function is the proximal operator $\Phi_{\frac{\lambda_1}{c}\mathbf{g}_l, \frac{\lambda_2}{c}\mathbf{g}_l, \hbar}(\mathbf{u})$ with learnable parameters $\lambda_1$, $\lambda_2$, $c$, $\mathbf{g}_l$ (see Fig. 1 for the shapes of the activation functions).

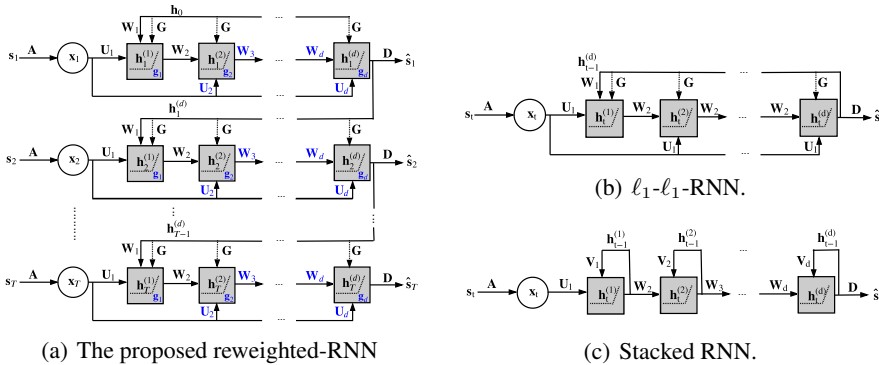

Figure 2: The proposed (a) reweighted-RNN vs. (b) $\ell_1$-$\ell_1$-RNN and (c) Stacked RNN with $d$ layers.

Fig. 2(a) depicts the architecture of the proposed reweighted-RNN. Input vectors $\mathbf{s}_t, t = 1, \ldots, T$ are compressed by a linear measurement layer $\mathbf{A}$, resulting in compressive measurements $\mathbf{x}_t$. The reconstructed vectors $\hat{\mathbf{s}}_t, t = 1, \ldots, T$, are obtained by multiplying linearly the hidden representation $\mathbf{h}_t^{(d)}$ with the dictionary $\mathbf{D}$. We train our network in an end-to-end fashion. During training, we minimize the loss function $\mathcal{L}(\boldsymbol{\Theta}) = \mathbb{E}_{\mathbf{s}_1, \cdots, \mathbf{s}_T} \left[ \sum_{t=1}^T \|\mathbf{s}_t - \hat{\mathbf{s}}_t\|_2^2 \right]$ using stochastic gradient descent on mini-batches, where the trainable parameters are $\boldsymbol{\Theta} = \{\mathbf{A}, \mathbf{D}, \mathbf{G}, \mathbf{h}_0, \mathbf{Z}_1, \ldots, \mathbf{Z}_d, \mathbf{g}_1, \ldots, \mathbf{g}_d, c, \lambda_1, \lambda_2\}$.

We now compare the proposed reweighted-RNN [Fig. 2(a)] against the recent $\ell_1$-$\ell_1$-RNN (Le et al., 2019) [Fig. 2(b)]. The $l$-th hidden layer in $\ell_1$-$\ell_1$-RNN is given by

$$\mathbf{h}_t^{(l)} = \begin{cases} \varPhi_{\frac{\lambda_1}{c}, \frac{\lambda_2}{c}, \mathbf{Gh}_{t-1}^{(d)}} \left( \mathbf{W}_1 \mathbf{h}_{t-1}^{(d)} + \mathbf{U}_1 \mathbf{x}_t \right), & \text{if } l = 1, \\ \varPhi_{\frac{\lambda_1}{c}, \frac{\lambda_2}{c}, \mathbf{Gh}_{t-1}^{(d)}} \left( \mathbf{W}_2 \mathbf{h}_t^{(l-1)} + \mathbf{U}_1 \mathbf{x}_t \right), & \text{if } l > 1. \end{cases} \tag{13}$$

The proposed model has the following advantages over $\ell_1$-$\ell_1$-RNN. Firstly, $\ell_1$-$\ell_1$-RNN uses the proximal operator $\varPhi_{\frac{\lambda_1}{c}, \frac{\lambda_2}{c}, \hbar}(\mathbf{u})$ as activation function, whose learnable parameters $\lambda_1$, $\lambda_2$ are fixed across the network. Conversely, the corresponding parameters $\frac{\lambda_1}{c}\mathbf{g}_l$ and $\frac{\lambda_2}{c}\mathbf{g}_l$ [see (7), (8), and Fig. 1] in our proximal operator, $\varPhi_{\frac{\lambda_1}{c}\mathbf{g}_l, \frac{\lambda_2}{c}\mathbf{g}_l, \hbar}(\mathbf{u})$, are learned for each hidden layer due to the reweighting vector $\mathbf{g}_l$; hence, the proposed model has a different activation function for each unit per layer. The second difference comes from the set of parameters $\{\mathbf{W}_l, \mathbf{U}_l\}$ in (13) and (9). The $\ell_1$-$\ell_1$-RNN model uses the same $\{\mathbf{W}_2, \mathbf{U}_1\}$ for the second and higher layers. In contrast, our reweighted-RNN has different sets of $\{\mathbf{W}_l, \mathbf{U}_l\}$ per hidden layer due to the reweighting matrix $\mathbf{Z}_l$. These two aspects [which are schematically highlighted in blue fonts in Fig. 2(a)] can lead to an increase in the learning capability of the proposed reweighted-RNN, especially when the depth of the model increases.

In comparison to a generic stacked RNN (Pascanu et al., 2014) [Fig. 2(c)], reweighted-RNN promotes the inherent data structure, that is, each vector $\mathbf{s}_t$ has a sparse representation $\mathbf{h}_t$ and consecutive $\mathbf{h}_t$'s are correlated. This design characteristic of the reweighted-RNN leads to residual connections which reduce the risk of vanishing gradients during training [the same idea has been shown in several works (He et al., 2016; Huang et al., 2017) in deep neural network literature]. Furthermore, in (10) and (12), we see a weight coupling of $\mathbf{W}_l$ and $\mathbf{U}_l$ (due to the shared components of $\mathbf{A}$, $\mathbf{D}$ and $\mathbf{Z}$). This coupling satisfies the necessary condition of the convergence in Chen et al. (2018) (Theorem 1). Using Theorem 2 in Chen et al. (2018), it can be shown that reweighted-RNN, in theory, needs a smaller number of iterations (i.e., $d$ in Algorithm 1) to reach convergence, compared to ISTA (Daubechies et al., 2004) and FISTA (Beck & Teboulle, 2009).

## 3 GENERALIZATION ERROR BOUND

While increasing the network expressivity, the over-parameterization of reweighted-RNN raises the question of whether our network ensures good generalization. In this section, we derive and analyze the generalization properties of the proposed reweighted-RNN model in comparison to state-of-the-art RNN architectures. We provide bounds on the Rademacher complexity (Shalev-Shwartz & Ben-David, 2014) for functional classes of the considered deep RNNs, which are used to derive generalization error bounds for evaluating their generalization properties (we refer to Appendix C.1 for definitions of the Rademacher complexity and the generalization error bound).

**Preliminaries:** We consider a deep RNN as a $d$-layer network $f_{\mathcal{W},\mathcal{U}}^{(d)} \in \mathcal{F}_{d,T} : \mathbb{R}^h \times \mathbb{R}^n \mapsto \mathbb{R}^h$ with weight parameters $\mathcal{W} = (\mathbf{W}_1, ..., \mathbf{W}_d)$ and $\mathcal{U} = (\mathbf{U}_1, ..., \mathbf{U}_d)$, where $\mathbf{W}_l \in \mathbb{R}^{h \times h}$, $\mathbf{U}_l \in \mathbb{R}^{h \times n}$. As in Bartlett et al. (2017); Golowich et al. (2018); Neyshabur et al. (2019; 2015), we derive generalization error bounds by controlling norms of the weight matrices. Let $\|\mathbf{W}_l\|_{p,q} = (\sum_j (\|\mathbf{w}_{l,j}\|_p)^q)^{1/q}$ define the $\ell_{p,q}$-norm, $p, q \geq 1$, of the weight-matrix $\mathbf{W}_l$, where $\mathbf{w}_{l,j} \in \mathbb{R}^h$ is the $j^{\text{th}}$ row of $\mathbf{W}_l$. Since we focus on deep networks with soft-thresholding-based activation units—designed by unfolding algorithms for $\ell_1$-norm minimization—we derive the network complexities under bounding per-unit $\ell_1$ regularization, i.e., $\|\mathbf{w}_{l,j}\|_1$. We also denote $\|\mathbf{W}_l\|_{1,\infty} = \max_j \|\mathbf{w}_{l,j}\|_1$ as the maximum of the $\ell_1$-norms of the matrix's rows. We assume that the $\ell_1$-norm of the weights of each neuron is bounded as $\|\mathbf{W}_l\|_{1,\infty} = \max_j \|\mathbf{w}_{l,j}\|_1 \leq \alpha_l$; similarly, $\|\mathbf{U}_l\|_{1,\infty} = \max_j \|\mathbf{u}_{l,j}\|_1 \leq \beta_l$, where $\mathbf{u}_{l,j}$ is the $j^{\text{th}}$ row of the matrix $\mathbf{U}_l$. As shown in (9), we can write the reweighted-RNN model recursively as $\mathbf{h}_t^{(1)} = f_{\mathcal{W},\mathcal{U}}^{(1)}(\mathbf{h}_{t-1}^{(d)}, \mathbf{x}_t) = \Phi(\mathbf{W}_1\mathbf{h}_{t-1}^{(d)} + \mathbf{U}_1\mathbf{x}_t)$ and $\mathbf{h}_t^{(l)} = f_{\mathcal{W},\mathcal{U}}^{(l)}(\mathbf{h}_{t-1}^{(d)}, \mathbf{x}_t) = \Phi(\mathbf{W}_l f_{\mathcal{W},\mathcal{U}}^{(l-1)}(\mathbf{h}_{t-1}^{(d)}, \mathbf{x}_t) + \mathbf{U}_l\mathbf{x}_t)$, where $\Phi(\cdot)$ is an activation function. For convenience, we denote the input layer as $f_{\mathcal{W},\mathcal{U}}^{(0)} = \mathbf{h}_{t-1}^{(d)}$; namely, at $t = 1$, we have $\mathbf{h}_0^{(l)} \equiv \mathbf{h}_0$.

We denote the true and training loss by $L_{\mathcal{D}}(f)$ and $L_S(f)$, respectively, where $S$ is the training set (of size $m$) drawn i.i.d. from the distribution $\mathcal{D}$. The generalization error is $L_{\mathcal{D}}(f) - L_S(f)$, with $f$ a function from the functional class $\mathcal{F}_{d,T}$. At time step $t$, we define $\mathbf{X}_t \in \mathbb{R}^{n \times m}$ as a matrix composed of $m$ columns from the input vectors $\{\mathbf{x}_{t,i}\}_{i=1}^m$. We also define $\|\mathbf{X}_t\|_{2,\infty} = \sqrt{\max_{k \in \{1,...,n\}} \sum_{i=1}^m \mathrm{x}_{t,i,k}^2}$ as the maximum of the $\ell_2$-norms of the rows of matrix $\mathbf{X}_t$, and $\|\mathbf{h}_0\|_\infty = \max_j |\mathrm{h}_{0,j}|$.

**Generalization error bound**. We first derive the generalization error bound for the proposed reweighted-RNN (with $T$ time steps) based on Rademacher complexity (see Theorem 26.5 in Shalev-Shwartz & Ben-David (2014) and Theorem C.1 in the Appendix).

**Theorem 3.1** (Generalization error bound). *Let $\mathcal{F}_{d,T} : \mathbb{R}^h \times \mathbb{R}^n \mapsto \mathbb{R}^h$ denote the functional class of reweighted-RNN with $T$ time steps and $d$ layers, where $\|\mathbf{W}_l\|_{1,\infty} \leq \alpha_l$, $\|\mathbf{U}_l\|_{1,\infty} \leq \beta_l$, and $1 \leq l \leq d$. Assume that the input data $\|\mathbf{X}_t\|_{2,\infty} \leq \sqrt{m}B_{\mathbf{x}}$, the initial hidden state is $\mathbf{h}_0$, and the loss function is 1-Lipschitz and bounded by $\eta$. Then, for $f \in \mathcal{F}_{d,T}$ and any $\delta > 0$, with probability at least $1 - \delta$ over a training set $S$ of size $m$ drawn i.i.d. from the distribution $\mathcal{D}$,*

$$L_{\mathcal{D}}(f) - L_S(f) \leq 2\mathfrak{R}_S(\mathcal{F}_{d,T}) + 4\eta\sqrt{\frac{2\log(4/\delta)}{m}}, \tag{14}$$

*where*

$$\mathfrak{R}_S(\mathcal{F}_{d,T}) \leq \sqrt{\frac{2(4dT\log 2 + \log n + \log h)}{m}} \cdot \sqrt{\Big(\sum_{l=1}^d \beta_l \Lambda_l\Big)^2 \Big(\frac{\Lambda_0^T - 1}{\Lambda_0 - 1}\Big)^2 B_{\mathbf{x}}^2 + \Lambda_0^{2T}\|\mathbf{h}_0\|_\infty^2}, \tag{15}$$

*with $\Lambda_l$ defined as follows: $\Lambda_l = \prod_{k=l+1}^d \alpha_k$ with $0 \leq l \leq d-1$ and $\Lambda_d = 1$.*

*Proof.* The proof is given in Appendix D. $\qquad\square$

The generalization error in (14) is bounded by the Rademacher complexity, which depends on the training set $S$. If the Rademacher complexity is small, the network can be learned with a small generalization error. The bound in (15) is in the order of the square root of the network depth $d$ multiplied by the number of time steps $T$. The bound depends on the logarithm of the number of measurements $n$ and the number of hidden units $h$. It is worth mentioning that the second square root in (15) only depends on the norm constraints and the input training data, and it is independent of the network depth $d$ and the number of time steps $T$ under the appropriate norm constraints.

To compare our model with $\ell_1$-$\ell_1$-RNN (Le et al., 2019) and Sista-RNN (Wisdom et al., 2017), we derive bounds on their Rademacher complexities for a time step $t$. The definitions of a functional class $\mathcal{F}_{d,t}$ for the $t^{th}$ time step of reweighted-RNN, $\ell_1$-$\ell_1$-RNN, and Sista-RNN are given in Appendix C.2. Let $\mathbf{H}_{t-1} \in \mathbb{R}^{h \times m}$ denote a matrix with columns the vectors of the previous hidden state $\{\mathbf{h}_{t-1,i}\}_{i=1}^m$, and $\|\mathbf{H}_{t-1}\|_{2,\infty} = \sqrt{\max_{k \in \{1,...,h\}} \sum_{i=1}^m \mathrm{h}_{t-1,i,k}^2} \leq \sqrt{m}B_{\mathbf{h}_{t-1}}$.

**Corollary 3.1.1.** *The empirical Rademacher complexity of $\mathcal{F}_{d,t}$ for reweighted-RNN is bounded as*

$$\Re_S(\mathcal{F}_{d,t}) \leq \sqrt{\frac{2(4d\log 2 + \log n + \log h)}{m}} \cdot \sqrt{\Big(\sum_{l=1}^{d} \beta_l \Lambda_l\Big)^2 B_{\mathbf{x}}^2 + \Lambda_0^2 B_{\mathbf{h}_{t-1}}^2}, \qquad (16)$$

*with $m$ the number of training samples and $\Lambda_l$ given by $\Lambda_d = 1$, $\Lambda_l = \prod\limits_{k=l+1}^{d} \alpha_k$ with $0 \leq l \leq d-1$.*

*Proof.* The proof is a special case of Theorem 3.1 for time step $t$. □

Following the proof of Theorem 3.1, we can obtain the Rademacher complexities for $\ell_1$-$\ell_1$-RNN and Sista-RNN:

**Corollary 3.1.2.** *The empirical Rademacher complexity of $\mathcal{F}_{d,t}$ for $\ell_1$-$\ell_1$-RNN is bounded as:*

$$\Re_S(\mathcal{F}_{d,t}) \leq \sqrt{\frac{2(4d\log 2 + \log n + \log h)}{m}} \cdot \sqrt{\beta_1^2 \Big(\frac{\alpha_2^d - 1}{\alpha_2 - 1}\Big)^2 B_{\mathbf{x}}^2 + \alpha_1^2 \alpha_2^{2(d-1)} B_{\mathbf{h}_{t-1}}^2}. \qquad (17)$$

**Corollary 3.1.3.** *The empirical Rademacher complexity of $\mathcal{F}_{d,t}$ for Sista-RNN is bounded as:*

$$\Re_S(\mathcal{F}_{d,t})$$

$$\leq \sqrt{\frac{2(4d\log 2 + \log n + \log h)}{m}} \cdot \sqrt{\beta_1^2 \Big(\frac{\alpha_2^d - 1}{\alpha_2 - 1}\Big)^2 B_{\mathbf{x}}^2 + \Big(\alpha_1 \alpha_2^{(d-1)} + \beta_2 \Big(\frac{\alpha_2^{d-1} - 1}{\alpha_2 - 1}\Big)\Big)^2 B_{\mathbf{h}_{t-1}}^2}. \qquad (18)$$

By contrasting (16) with (17) and (18), we see that the complexities of $\ell_1$-$\ell_1$-RNN and Sista-RNN have a polynomial dependence on $\alpha_1$, $\beta_1$ and $\alpha_2$, $\beta_2$ (the norms of first two layers), whereas the complexity of reweighted-RNN has a polynomial dependence on $\alpha_1, \ldots, \alpha_d$ and $\beta_1, \ldots, \beta_d$ (the norms of all layers). This over-parameterization offers a flexible way to control the generalization error of reweighted-RNN. We derive empirical generalization errors in Fig. 6 in Appendix A demonstrating that increasing the depth of reweighted-RNN still ensures the low generalization error.

## 4 EXPERIMENTAL RESULTS

We assess the proposed RNN model in the task of video-frame reconstruction from compressive measurements. The performance is measured using the peak signal-to-noise ratio (PSNR) between the reconstructed $\hat{\mathbf{s}}_t$ and the original frame $\mathbf{s}_t$. We use the moving MNIST dataset (Srivastava et al., 2015), which contains 10K video sequences of equal length (20 frames per sequence). Similar to the setup in Le et al. (2019), the dataset is split into training, validation, and test sets of 8K, 1K, and 1K sequences, respectively. In order to reduce the training time and memory requirements, we downscale the frames from $64 \times 64$ to $16 \times 16$ pixels using bilinear decimation. After vectorizing, we obtain sequences of $\mathbf{s}_1, \ldots, \mathbf{s}_T \in \mathbb{R}^{256}$. Per sequence, we obtain measurements $\mathbf{x}_1, \ldots, \mathbf{x}_T \in \mathbb{R}^n$ using a trainable linear sensing matrix $\mathbf{A} \in \mathbb{R}^{n \times n_0}$, with $T = 20$, $n_0 = 256$ and $n < n_0$.

We compare the reconstruction performance of the proposed reweighted-RNN model against deep-unfolding RNN models, namely, $\ell_1$-$\ell_1$-RNN (Le et al., 2019), Sista-RNN (Wisdom et al., 2017), and stacked-RNN models, that is, sRNN (Elman, 1990), LSTM (Hochreiter & Schmidhuber, 1997), GRU (Cho et al., 2014), FastRNN (Kusupati et al., 2018)[1], IndRNN (Li et al., 2018) and SpectralRNN (Zhang et al., 2018). For the vanilla RNN, LSTM and GRU, the native Pytorch cell implementations were used. The unfolding-based methods were implemented in Pytorch[2], with Sista-RNN and $\ell_1$-$\ell_1$-RNN tested by reproducing the experiments in Wisdom et al. (2017); Le et al. (2019). For FastRNN, IndRNN, and SpectralRNN cells, we use the publically available Tensorflow implementations. While Sista-RNN, $\ell_1$-$\ell_1$-RNN and reweighted-RNN have their own layer-stacking schemes derived from unfolding minimization algorithms, we use the stacking rule in Pascanu et al. (2013) [see Fig 2(c)] to build deep networks for other RNN architectures.

---

[1]Kusupati et al. (2018) also proposed FastGRNN; we found that, in our application scenario, the non-gated variant (the FastRNN) consistently outperformed FastGRNN. As such, we do not include results with the latter.

[2]Our implementations are available at https://1drv.ms/u/s!ApHn770BvhH2aWay9xEhAiXydfo?e=aCX1X0.

Our default settings are: a compressed sensing (CS) rate of $n/n_0 = 0.2$, $d = 3$ hidden layers[3] with $h = 2^{10}$ hidden units per layer. In each set of experiments, we vary each of these hyper-parameters while keeping the other two fixed. For the unfolding methods, the overcomplete dictionary $\mathbf{D} \in \mathbb{R}^{n_0 \times h}$ is initialized with the discrete cosine transform (DCT) with varying dictionary sizes of $h = \{2^7, 2^8, 2^9, 2^{10}, 2^{11}, 2^{12}\}$ (corresponding to a number of hidden neurons in the other methods). For initializing $\lambda_1, \lambda_2$ [see (2), (4)], we perform a random search in the range of $[10^{-5}, 3.0]$ in the validation set. To avoid the problem of exploding gradients, we clip the gradients during backpropagation such that the $\ell_2$-norms are less than or equal to 0.25. We do not apply weight decay regularization as we found it often leads to worse performance, especially since gradient clipping is already used for training stability. We train the networks for 200 epochs using the Adam optimizer with an initial learning rate of 0.0003, and a batch size of 32. During training, if the validation loss does not decrease for 5 epochs, we reduce the learning rate to 0.3 of its current value.

Table 1 summarizes the reconstruction results for different CS rates $n/n_0$. The reweighted-RNN model systematically outperforms the other models, often by a large margin. Table 2 shows similar improvements for various dimensions of hidden units. Table 3 shows that IndRNN delivers higher reconstruction performance than our model when a small number of hidden layers ($d = 1, 2$) is used. Moreover, when the depth increases, reweighted-RNN surpasses all other models. Our network also has fewer trainable parameters compared to the popular variants of RNN. At the default settings, reweighted-RNN, the stacked vanilla RNN, the stacked LSTM, and the stacked GRU have 4.47M, 5.58M, 21.48M, and 16.18M parameters, respectively.

Table 1: Average PSNR [dB] on the test set with different CS rates.

| CS Rate | sRNN | LSTM | GRU | IndRNN | FastRNN | SpectralRNN | Sista-RNN | $\ell_1$-$\ell_1$-RNN | Ours |
|---|---|---|---|---|---|---|---|---|---|
| 0.1 | 25.11 | 24.58 | 25.18 | 25.68 | 25.21 | 25.15 | 25.16 | 24.68 | **26.25** |
| 0.2 | 31.14 | 29.46 | 31.19 | 32.90 | 32.05 | 31.65 | 31.53 | 30.79 | **34.19** |
| 0.3 | 35.38 | 32.91 | 36.49 | 37.12 | 36.40 | 36.89 | 36.96 | 37.77 | **42.39** |
| 0.4 | 38.05 | 34.95 | 39.47 | 40.84 | 39.21 | 40.22 | 39.57 | 40.35 | **46.03** |
| 0.5 | 39.34 | 36.28 | 41.12 | 45.49 | 41.87 | 41.36 | 41.56 | 43.35 | **48.70** |

Table 2: Average PSNR [dB] on the test set with different network widths $h$ ($\frac{n}{n_0} = 0.2$, $d = 3$).

| $h$ | sRNN | LSTM | GRU | IndRNN | FastRNN | SpectralRNN | Sista-RNN | $\ell_1$-$\ell_1$-RNN | Ours |
|---|---|---|---|---|---|---|---|---|---|
| $2^7$ | 23.35 | 22.87 | 23.55 | 23.82 | 23.83 | 22.92 | 23.86 | 23.90 | **28.09** |
| $2^8$ | 25.81 | 23.88 | 26.67 | 27.10 | 26.71 | 24.46 | 29.64 | 29.55 | **31.46** |
| $2^9$ | 28.72 | 26.83 | 30.29 | 32.03 | 29.92 | 30.23 | 31.30 | 30.61 | **33.61** |
| $2^{10}$ | 31.14 | 29.46 | 31.19 | 32.90 | 32.05 | 31.65 | 31.53 | 30.79 | **34.19** |
| $2^{11}$ | 29.91 | 29.30 | 31.15 | 33.10 | 30.80 | 31.68 | 31.82 | 30.45 | **34.80** |
| $2^{12}$ | 29.71 | 29.08 | 30.93 | 32.47 | 24.26 | 29.26 | 31.63 | 30.09 | **34.98** |

Table 3: Average PSNR [dB] on the test set with different network depths $d$ ($\frac{n}{n_0} = 0.2$, $h = 2^{10}$).

| $d$ | sRNN | LSTM | GRU | IndRNN | FastRNN | SpectralRNN | Sista-RNN | $\ell_1$-$\ell_1$-RNN | Ours |
|---|---|---|---|---|---|---|---|---|---|
| 1 | 27.52 | 27.76 | 27.61 | **30.12** | 29.32 | 29.62 | 28.41 | 28.49 | 29.19 |
| 2 | 29.21 | 29.46 | 29.68 | **32.73** | 30.84 | 31.37 | 30.67 | 30.19 | 32.12 |
| 3 | 31.14 | 22.29 | 31.19 | 32.90 | 32.05 | 31.65 | 31.53 | 30.79 | **34.19** |
| 4 | 31.64 | 16.50 | 29.26 | 20.65 | 31.07 | 31.10 | 32.56 | 31.80 | **35.99** |
| 5 | 16.50 | 26.66 | 16.50 | 25.17 | 20.10 | 30.52 | 33.07 | 32.50 | **36.91** |
| 6 | 22.28 | 16.50 | 16.50 | 20.90 | 19.37 | 29.56 | 31.99 | 32.00 | **38.90** |

## 5 CONCLUSIONS

We designed a novel deep RNN by unfolding an algorithm that solves a reweighted $\ell_1$-$\ell_1$ minimization problem. Our model has high network expressivity due to *per-unit learnable activation functions* and *over-parameterized weights*. We also established the generalization error bound for the proposed model via Rademacher complexity. We showed that reweighted-RNN has good generalization properties and its error bound is tighter than existing RNNs in function of the number of time steps. Experimentation on the task of sequential video-frame reconstruction shows that our model (*i*) outperforms various state-of-the-art RNNs in terms of accuracy and (*ii*) is capable of stacking many hidden layers resulting in a better learning capability than the existing unfolding methods.

---

[3]In our experiments, the 2-layer LSTM network outperforms the 3-layer one (see Table 3), the default setting for LSTM is thus using 2 layers.

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

# A    SUPPLEMENTARY EXPERIMENTS

In our experiments, we use the publically available Tensorflow implementations for FastRNN[4], IndRNN[5], and SpectralRNN[6] cells. While Sista-RNN, $\ell_1$-$\ell_1$-RNN and reweighted-RNN have their own layer-stacking schemes derived from unfolding minimization algorithms, we use the stacking rule in Pascanu et al. (2013) [see Fig 2(c)] to build deep networks for other RNN models.

Figure 3 shows the learning curves of all methods under the default setting. It can be seen that reweighted-RNN achieves the lowest mean square error on both the training and validation sets. It can also be observed that the unfolding methods converge faster than the stacked RNNs, with the proposed reweighted-RNN being the fastest. More experimental results for the proposed reweighted-RNN are provided to illustrate the learning curves, which measure the average mean square error vs. the training epochs between the original frames and their reconstructed counterparts, with different CS rates [Fig. 4], different network depths $d$ [Fig. 6], and different network widths $h$ [Fig. 5].

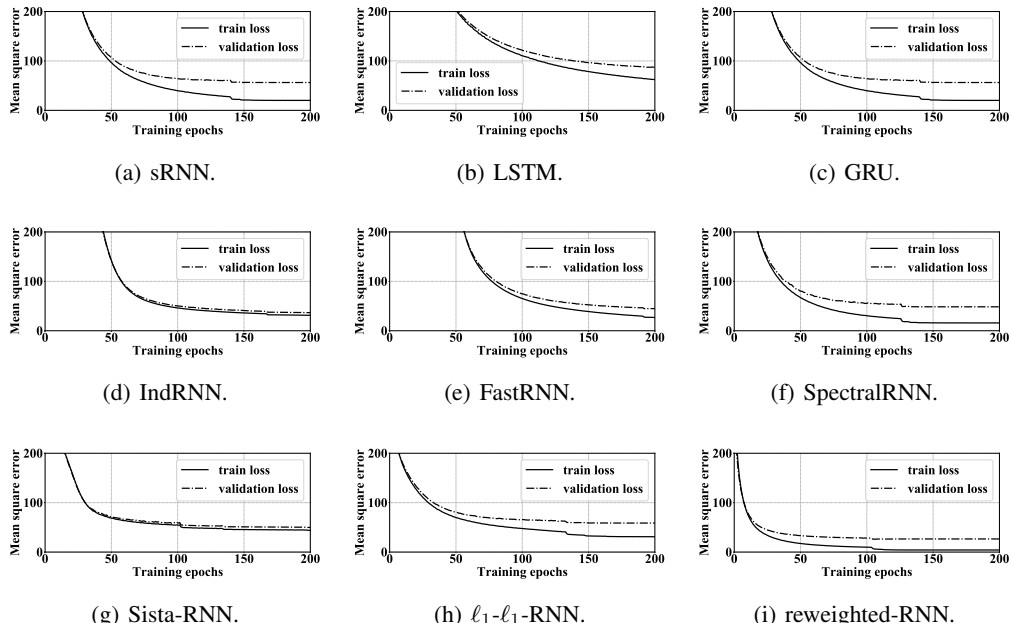

(a) sRNN.                    (b) LSTM.                    (c) GRU.

(d) IndRNN.                  (e) FastRNN.                 (f) SpectralRNN.

(g) Sista-RNN.              (h) $\ell_1$-$\ell_1$-RNN.              (i) reweighted-RNN.

Figure 3: Average mean square error between the original and reconstructed frames vs. training epoches on the training and the validation sets for the default setting (a CS rate is $0.2$, $d = 3$, $h = 2^{10}$).

Since we use different frameworks to implement the RNNs used in our benchmarks, we do not report and compare the computational time for training of the models. Specifically, we rely on the Tensorflow implementations from the authors of Independent-RNN, Fast-RNN and Spectral RNN, while the rest is written in Pytorch. Furthermore, even among Pytorch models, the vanilla RNN, LSTM, and GRU cells are written in CuDNN (default Pytorch implementations), so that they are significantly faster in training than the others. This does not mean that these networks have better runtime complexities, but rather more efficient implementations. However, an important comparison could be made between $\ell_1$-$\ell_1$-RNN Le et al. (2019) (as the baseline method) and Reweighted-RNN due to their similarities in implementations. At the default settings, it takes 3,521 seconds and 2,985 seconds to train Reweighted-RNN and $\ell_1$-$\ell_1$-RNN Le et al. (2019), respectively.

---

[4]Code available at https://github.com/microsoft/EdgeML

[5]Code available at https://github.com/batzner/indrnn

[6]Code available at https://github.com/zhangjiong724/spectral-RNN

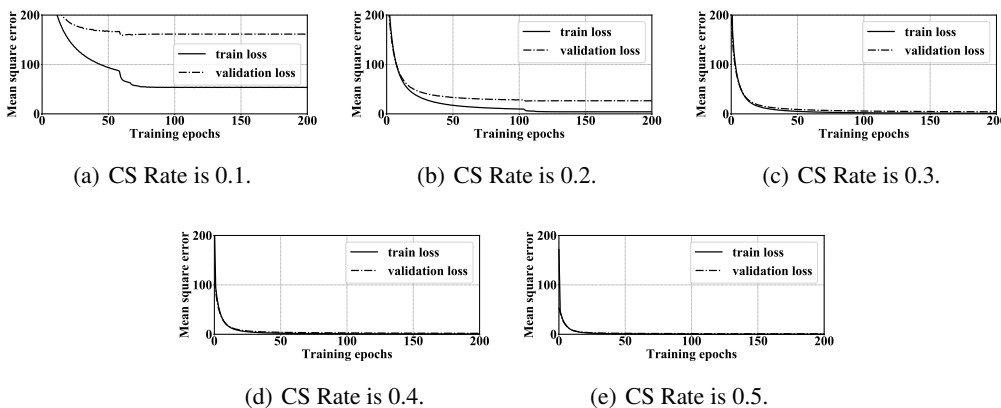

Figure 4: Reweighted-RNN: Average mean square error between the original and reconstructed frames vs. training epoches on the training and the validation sets with different CS rates ($d = 3$, $h = 2^{10}$).

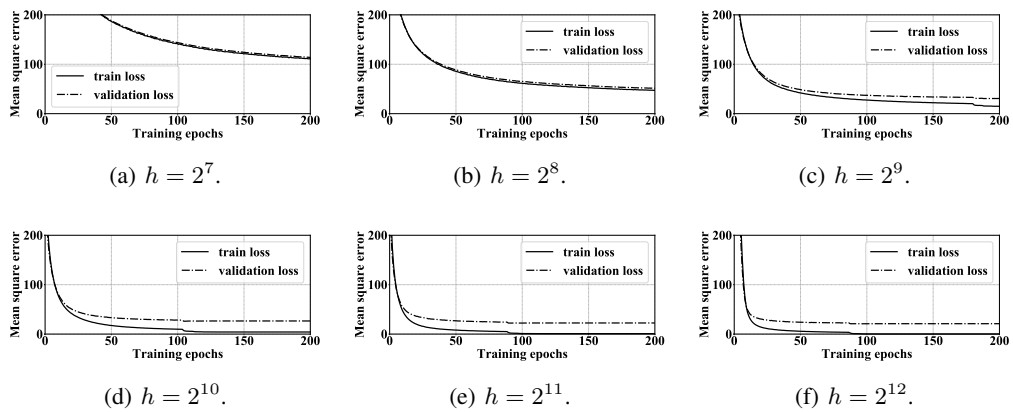

Figure 5: Reweighted-RNN: Average mean square error between the original and reconstructed frames vs. training epoches on the training and the validation sets with different network widths $h$ (a CS rates is 0.2, $d = 3$).

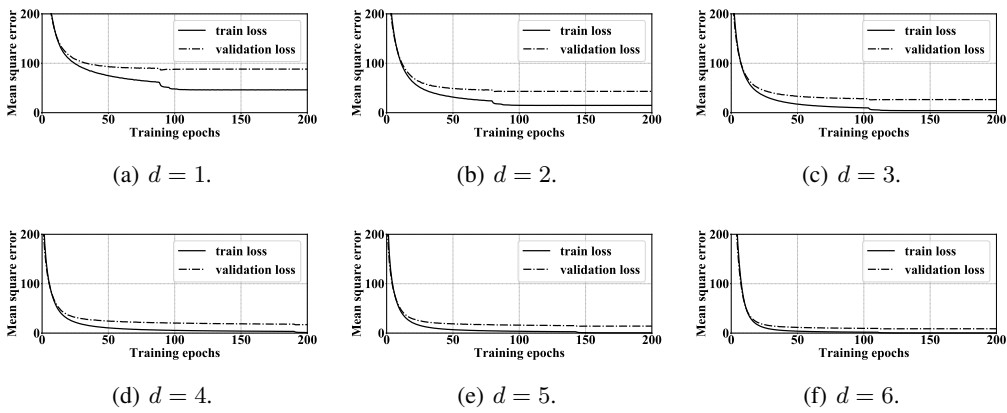

Figure 6: Reweighted-RNN: Average mean square error between the original and reconstructed frames vs. training epoches on the training and the validation sets with different network depths $d$ (a CS rate is 0.2, $h = 2^{10}$).

## B   THE PROXIMAL OPERATOR FOR PROBLEM (4)

**Proposition B.1.** *The proximal operator* $\Phi_{\frac{\lambda_1}{c}\mathrm{g},\frac{\lambda_2}{c}\mathrm{g},\hbar}(u)$ *in (6) for the reweighted* $\ell_1$-$\ell_1$ *minimization problem (4), for which* $g(v) = \lambda_1\mathrm{g}|v| + \lambda_2\mathrm{g}|v - \hbar|$, *is given by*

$$\Phi_{\frac{\lambda_1}{c}\mathrm{g},\frac{\lambda_2}{c}\mathrm{g},\hbar\geq0}(u) = \begin{cases} u - \frac{\lambda_1}{c}\mathrm{g} - \frac{\lambda_2}{c}\mathrm{g}, & \hbar + \frac{\lambda_1}{c}\mathrm{g} + \frac{\lambda_2}{c}\mathrm{g} < u < \infty \\ \hbar, & \hbar + \frac{\lambda_1}{c}\mathrm{g} - \frac{\lambda_2}{c}\mathrm{g} \leq u \leq \hbar + \frac{\lambda_1}{c}\mathrm{g} + \frac{\lambda_2}{c}\mathrm{g} \\ u - \frac{\lambda_1}{c}\mathrm{g} + \frac{\lambda_2}{c}\mathrm{g}, & \frac{\lambda_1}{c}\mathrm{g} - \frac{\lambda_2}{c}\mathrm{g} < u < \hbar + \frac{\lambda_1}{c}\mathrm{g} - \frac{\lambda_2}{c}\mathrm{g} \\ 0, & -\frac{\lambda_1}{c}\mathrm{g} - \frac{\lambda_2}{c}\mathrm{g} \leq u \leq \frac{\lambda_1}{c}\mathrm{g} - \frac{\lambda_2}{c}\mathrm{g} \\ u + \frac{\lambda_1}{c}\mathrm{g} + \frac{\lambda_2}{c}\mathrm{g}, & -\infty < u < -\frac{\lambda_1}{c}\mathrm{g} - \frac{\lambda_2}{c}\mathrm{g} \end{cases} \tag{19}$$

$$\Phi_{\frac{\lambda_1}{c}\mathrm{g},\frac{\lambda_2}{c}\mathrm{g},\hbar<0}(u) = \begin{cases} u - \frac{\lambda_1}{c}\mathrm{g} - \frac{\lambda_2}{c}\mathrm{g}, & \frac{\lambda_1}{c}\mathrm{g} + \frac{\lambda_2}{c}\mathrm{g} < u < \infty \\ 0, & -\frac{\lambda_1}{c}\mathrm{g} + \frac{\lambda_2}{c}\mathrm{g} \leq u \leq \frac{\lambda_1}{c}\mathrm{g} + \frac{\lambda_2}{c}\mathrm{g} \\ u + \frac{\lambda_1}{c}\mathrm{g} - \frac{\lambda_2}{c}\mathrm{g}, & \hbar - \frac{\lambda_1}{c}\mathrm{g} + \frac{\lambda_2}{c}\mathrm{g} < u < -\frac{\lambda_1}{c}\mathrm{g} + \frac{\lambda_2}{c}\mathrm{g} \\ \hbar, & \hbar - \frac{\lambda_1}{c}\mathrm{g} - \frac{\lambda_2}{c}\mathrm{g} \leq u \leq \hbar - \frac{\lambda_1}{c}\mathrm{g} + \frac{\lambda_2}{c}\mathrm{g} \\ u - \frac{\lambda_1}{c}\mathrm{g} + \frac{\lambda_2}{c}\mathrm{g}, & -\infty < u < \hbar - \frac{\lambda_1}{c}\mathrm{g} - \frac{\lambda_2}{c}\mathrm{g} \end{cases} \tag{20}$$

*Proof.* We compute the proximal operator $\Phi_{\frac{\lambda_1}{c}\mathrm{g},\frac{\lambda_2}{c}\mathrm{g},\hbar}(u)$ (19) for $\hbar \geq 0$, it is similar for $\hbar < 0$. From (6), $\Phi_{\frac{\lambda_1}{c}\mathrm{g},\frac{\lambda_2}{c}\mathrm{g},\hbar}(u)$ is expressed by:

$$\Phi_{\frac{\lambda_1}{c}\mathrm{g},\frac{\lambda_2}{c}\mathrm{g},\hbar}(u) = \arg\min_{v\in\mathbb{R}} \left\{ \varphi(v) := \frac{\lambda_1}{c}\mathrm{g}|v| + \frac{\lambda_2}{c}\mathrm{g}|v - \hbar| + \frac{1}{2}|v - u|^2 \right\}. \tag{21}$$

We first consider the $\partial\varphi(v)/\partial v$ in $v \in \{(-\infty, 0), (0, \hbar), (\hbar, \infty)\}$, in which $\partial\varphi(v)$ exists. Taking the derivative of $\varphi(v)$ in these intervals delivers

$$\frac{\partial\varphi(v)}{\partial v} = \frac{\lambda_1}{c}\mathrm{g} \cdot \mathrm{sign}(v) + \frac{\lambda_2}{c}\mathrm{g} \cdot \mathrm{sign}(v - \hbar) + (v - u), \tag{22}$$

where $\mathrm{sign}(.)$ is a sign function. When setting $\partial\varphi(v)/\partial v = 0$ to minimize $\varphi(v)$, we derive:

$$v = \begin{cases} u - \frac{\lambda_1}{c}\mathrm{g} - \frac{\lambda_2}{c}\mathrm{g}, & \hbar < v < \infty \\ u - \frac{\lambda_1}{c}\mathrm{g} + \frac{\lambda_2}{c}\mathrm{g}, & 0 < v < \hbar \\ u + \frac{\lambda_1}{c}\mathrm{g} + \frac{\lambda_2}{c}\mathrm{g}, & -\infty < v < 0 \end{cases} \tag{23}$$

From (21) and (23), we have

$$\Phi_{\frac{\lambda_1}{c}\mathrm{g},\frac{\lambda_2}{c}\mathrm{g},\hbar}(u) = \begin{cases} u - \frac{\lambda_1}{c}\mathrm{g} - \frac{\lambda_2}{c}\mathrm{g}, & \hbar + \frac{\lambda_1}{c}\mathrm{g} + \frac{\lambda_2}{c}\mathrm{g} < u < \infty \\ u - \frac{\lambda_1}{c}\mathrm{g} + \frac{\lambda_2}{c}\mathrm{g}, & \frac{\lambda_1}{c}\mathrm{g} - \frac{\lambda_2}{c}\mathrm{g} < u < \hbar + \frac{\lambda_1}{c}\mathrm{g} - \frac{\lambda_2}{c}\mathrm{g} \\ u + \frac{\lambda_1}{c}\mathrm{g} + \frac{\lambda_2}{c}\mathrm{g}, & -\infty < u < -\frac{\lambda_1}{c}\mathrm{g} - \frac{\lambda_2}{c}\mathrm{g} \end{cases} \tag{24}$$

In the remaining range value of $u$, namely, $-\frac{\lambda_1}{c}\mathrm{g} - \frac{\lambda_2}{c}\mathrm{g} \leq u \leq \frac{\lambda_1}{c}\mathrm{g} - \frac{\lambda_2}{c}\mathrm{g}$ and $\hbar + \frac{\lambda_1}{c}\mathrm{g} - \frac{\lambda_2}{c}\mathrm{g} \leq u \leq \hbar + \frac{\lambda_1}{c}\mathrm{g} + \frac{\lambda_2}{c}\mathrm{g}$, we prove that the minimum of $\varphi(v)$ (21) is obtained when $v = 0$ and $v = \hbar$, respectively.

Let us rewrite $\varphi(v)$, which was defined in (21), as

$$\varphi(v) = \frac{\lambda_1}{c}\mathrm{g}|v| + \frac{\lambda_2}{c}\mathrm{g}|v - \hbar| + \frac{1}{2}|v - u|^2 \tag{25}$$

By applying the inequality $|a - b| \geq |a| - |b|$, where $a, b \in \mathbb{R}$, to (25), we obtain:

$$\varphi(v) \geq \frac{\lambda_1}{c}\mathrm{g}|v| + \frac{\lambda_2}{c}\mathrm{g}|v| - \frac{\lambda_2}{c}\mathrm{g}|\hbar| + \frac{1}{2}v^2 - vu + \frac{1}{2}u^2$$

$$\geq |v|\left(\frac{\lambda_1}{c}\mathrm{g} + \frac{\lambda_2}{c}\mathrm{g} - |u|\right) + \frac{1}{2}v^2 - \frac{\lambda_2}{c}\mathrm{g}|\hbar| + \frac{1}{2}u^2 \tag{26}$$

For $-\frac{\lambda_1}{c}\mathrm{g} - \frac{\lambda_2}{c}\mathrm{g} \leq u \leq \frac{\lambda_1}{c}\mathrm{g} - \frac{\lambda_2}{c}\mathrm{g}$, from (26), $\varphi(v)$ is minimal when $v = 0$, due to $\frac{\lambda_1}{c}\mathrm{g} + \frac{\lambda_2}{c}\mathrm{g} - |u| \geq 0$.

Similarly, for $\hbar + \frac{\lambda_1}{c}\mathbf{g} - \frac{\lambda_2}{c}\mathbf{g} \leq u \leq \hbar + \frac{\lambda_1}{c}\mathbf{g} + \frac{\lambda_2}{c}\mathbf{g}$, i.e., $\frac{\lambda_1}{c}\mathbf{g} - \frac{\lambda_2}{c}\mathbf{g} \leq u - \hbar \leq \frac{\lambda_1}{c}\mathbf{g} + \frac{\lambda_2}{c}\mathbf{g}$, we have

$$\varphi(v) \geq \frac{\lambda_1}{c}\mathbf{g}|v - \hbar| - \frac{\lambda_1}{c}\mathbf{g}|\hbar| + \frac{\lambda_2}{c}\mathbf{g}|v - \hbar| + \frac{1}{2}(v - \hbar)^2 - |v - \hbar||u - \hbar| + \frac{1}{2}(u - \hbar)^2$$

$$\geq |v - \hbar|\Big(\frac{\lambda_1}{c}\mathbf{g} + \frac{\lambda_2}{c}\mathbf{g} - |u - \hbar|\Big) + \frac{1}{2}(v - \hbar)^2 - \frac{\lambda_1}{c}\mathbf{g}|\hbar| + \frac{1}{2}(u - \hbar)^2. \tag{27}$$

From (27), $\varphi(v)$ is minimal when $v = \hbar$, since $\frac{\lambda_1}{c}\mathbf{g} + \frac{\lambda_2}{c}\mathbf{g} - |u - \hbar| \geq 0$. Combining these results with the result in (24), we conclude the proof. $\qquad\square$

## C  GENERALIZATION AND DEEP UNFOLDING RNNS

### C.1  GENERALIZATION ERROR BOUND DEFINITION

**Notations**. Let $f_{\mathcal{W}}^{(d)} : \mathbb{R}^n \mapsto \mathbb{R}^h$ be the function computed by a $d$-layer network with weight parameters $\mathcal{W}$. The network $f_{\mathcal{W}}^{(d)}$ maps an input sample $\mathbf{x}_i \in \mathbb{R}^n$ (from an input space $\mathcal{X}$) to an output $\mathbf{y}_i \in \mathbb{R}^h$ (from an output space $\mathcal{Y}$), i.e., $\mathbf{y}_i = f_{\mathcal{W}}^{(d)}(\mathbf{x}_i)$. Let $S$ denote a training set of size $m$, i.e., $S = \{(\mathbf{x}_i, \mathbf{y}_i)\}_{i=1}^m$ and $\mathbb{E}_{(\mathbf{x}_i, \mathbf{y}_i) \sim S}[\cdot]$ denote an expectation over $(\mathbf{x}_i, \mathbf{y}_i)$ from $S$. The set $S$ is drawn i.i.d. from a distribution $\mathcal{D}$, denoted as $S \sim \mathcal{D}^m$, over a space $\mathcal{Z} = \mathcal{X} \times \mathcal{Y}$. Let $\mathcal{F}$ be a (class) set of functions. Let $\ell : \mathcal{F} \times \mathcal{Z} \mapsto \mathbb{R}$ denote the loss function and $\ell \circ \mathcal{F} = \{z \mapsto \ell(f, z) : f \in \mathcal{F}\}$. We define the true loss and the empirical (training) loss by $L_{\mathcal{D}}(f)$ and $L_S(f)$, respectively, as follows:

$$L_{\mathcal{D}}(f) = \mathbb{E}_{(\mathbf{x}_i, \mathbf{y}_i) \sim \mathcal{D}}\big[\ell\big(f(\mathbf{x}_i), \mathbf{y}_i\big)\big], \tag{28}$$

and

$$L_S(f) = \mathbb{E}_{(\mathbf{x}_i, \mathbf{y}_i) \sim S}\big[\ell\big(f(\mathbf{x}_i), \mathbf{y}_i\big)\big]. \tag{29}$$

The *generalization error*, which is defined as a measure of how accurately a learned algorithm is able to predict outcome values for unseen data, is calculated by $L_{\mathcal{D}}(f) - L_S(f)$.

**Rademacher complexity**. Let $\mathcal{F}$ be a hypothesis set of functions (neural networks). The *empirical Rademacher complexity* of $\mathcal{F}$ (Shalev-Shwartz & Ben-David, 2014) for a training sample set $S$ is defined as follows:

$$\mathfrak{R}_S(\mathcal{F}) = \frac{1}{m} \mathbb{E}_{\boldsymbol{\epsilon} \in \{\pm 1\}^m} \left[ \sup_{f \in \mathcal{F}} \sum_{i=1}^m \epsilon_i f(\mathbf{x}_i) \right], \tag{30}$$

where $\boldsymbol{\epsilon} = (\epsilon_1, ..., \epsilon_m)$; here $\epsilon_i$ are independent uniformly distributed random (Rademacher) variables from $\{\pm 1\}$, according to $\mathbb{P}[\epsilon_i = 1] = \mathbb{P}[\epsilon_i = -1] = 1/2$.

**The generalization error bound** (Shalev-Shwartz & Ben-David, 2014) is derived based on the Rademacher complexity defined in the following theorem:

**Theorem C.1.** *(Shalev-Shwartz & Ben-David, 2014, Theorem 26.5)*
*Assume that $|\ell(f, z)| \leq \eta$ for all $f \in \mathcal{F}$ and $z$. Then, for any $\delta > 0$, with probability at least $1 - \delta$,*

$$L_{\mathcal{D}}(f) - L_S(f) \leq 2\mathfrak{R}_S(\ell \circ \mathcal{F}) + 4\eta\sqrt{\frac{2\log(4/\delta)}{m}}. \tag{31}$$

It can be noted that the bound in (31) via the Rademacher complexity depends on the training set $S$, which makes it applicable to a number of learning problems, e.g., regression and classification, under given a loss function $\ell$.

### C.2  NOTATION FOR DEEP UNFOLDED RNNS

In this subsection, we provide the required notation for the proposed reweighted-RNN model, $\ell_1$-$\ell_1$-RNN, and Sista-RNN, which will be used in the derivation of their respective generalization analysis.

**The proposed reweighted-RNN**. Let $\mathbf{h}_t^{(l)}$ be the hidden states in layer $l$ evolving in time step $t$. We write the reweighted-RNN model recursively as $\mathbf{h}_t^{(1)} = f_{\mathcal{W},\mathcal{U}}^{(1)}(\mathbf{h}_{t-1}^{(d)}, \mathbf{x}_t) = \Phi(\mathbf{W}_1\mathbf{h}_{t-1}^{(d)} + \mathbf{U}_1\mathbf{x}_t)$ and $\mathbf{h}_t^{(l)} = f_{\mathcal{W},\mathcal{U}}^{(l)}(\mathbf{h}_{t-1}^{(d)}, \mathbf{x}_t) = \Phi\Big(\mathbf{W}_l f_{\mathcal{W},\mathcal{U}}^{(l-1)}(\mathbf{h}_{t-1}^{(d)}, \mathbf{x}_t) + \mathbf{U}_l\mathbf{x}_t\Big)$, where $\Phi$ is an activation function. The

hidden state is updated as shown in (9). The real-valued family of functions, $\mathcal{F}_{d,t} : \mathbb{R}^h \times \mathbb{R}^n \mapsto \mathbb{R}$, for the functions $f_{\mathcal{W},\mathbf{U}}^{(d)}$ in layer $d$ is defined by:

$$\mathcal{F}_{d,t} = \left\{ (\mathbf{h}_{t-1}^{(d)}, \mathbf{x}_t) \mapsto \varPhi(\mathbf{w}_d^{\mathrm{T}} f_{\mathcal{W},\mathcal{U}}^{(d-1)}(\mathbf{h}_{t-1}^{(d)}, \mathbf{x}_t) + \mathbf{u}_d^{\mathrm{T}}\mathbf{x}_t) : \|\mathbf{W}_d\|_{1,\infty} \leq \alpha_d, \ \|\mathbf{U}_d\|_{1,\infty} \leq \beta_d \right\}, \tag{32}$$

where $\alpha_l, \beta_l$ are nonnegative hyper-parameters for layer $l$, where $1 < l \leq d$. In layer $l = 1$, the real-valued family of functions, $\mathcal{F}_{1,t} : \mathbb{R}^h \times \mathbb{R}^n \mapsto \mathbb{R}$, for the functions $f_{\mathcal{W},\mathcal{U}}^{(1)}$ is defined by:

$$\mathcal{F}_{1,t} = \left\{ (\mathbf{h}_{t-1}^{(d)}, \mathbf{x}_t) \mapsto \varPhi(\mathbf{w}_1^{\mathrm{T}}\mathbf{h}_{t-1}^{(d)} + \mathbf{u}_1^{\mathrm{T}}\mathbf{x}_t) : \|\mathbf{W}_1\|_{1,\infty} \leq \alpha_1, \ \|\mathbf{U}\|_{1,\infty} \leq \beta_1 \right\}, \tag{33}$$

where $\alpha_1, \beta_1$ are nonnegative hyper-parameters. We denote the input layer as $f_{\mathcal{W},\mathcal{U}}^{(0)} = \mathbf{h}_{t-1}^{(d)}$, in particular, at $t = 1$, $\mathbf{h}_0^{(l)} \equiv \mathbf{h}_0$.

**The $\ell_1$-$\ell_1$-RNN model** (Le et al., 2019). The hidden state $\mathbf{h}_t^{(l)}$ for $\ell_1$-$\ell_1$-RNN is updated as shown in (13). The real-valued family of functions, $\mathcal{F}_{d,t} : \mathbb{R}^h \times \mathbb{R}^n \mapsto \mathbb{R}$, for the function $f_{\mathcal{W},\mathbf{U}}^{(d)}$ in layer $d$ is defined by:

$$\mathcal{F}_{d,t} = \left\{ (\mathbf{h}_{t-1}^{(d)}, \mathbf{x}_t) \mapsto \varPhi(\mathbf{w}_2^{\mathrm{T}} f_{\mathcal{W},\mathcal{U}}^{(d-1)}(\mathbf{h}_{t-1}^{(d)}, \mathbf{x}_t) + \mathbf{u}_1^{\mathrm{T}}\mathbf{x}_t) : \|\mathbf{W}_2\|_{1,\infty} \leq \alpha_2, \ \|\mathbf{U}_1\|_{1,\infty} \leq \beta_1 \right\}, \tag{34}$$

where $\alpha_2, \beta_1$ are nonnegative hyper-parameters for layer $l$, where $1 < l \leq d$. In layer $l = 1$, the real-valued family of functions, $\mathcal{F}_{1,t} : \mathbb{R}^h \times \mathbb{R}^n \mapsto \mathbb{R}$, for the functions $f_{\mathcal{W},\mathcal{U}}^{(1)}$ is defined by:

$$\mathcal{F}_{1,t} = \left\{ (\mathbf{h}_{t-1}^{(d)}, \mathbf{x}_t) \mapsto \varPhi(\mathbf{w}_1^{\mathrm{T}}\mathbf{h}_{t-1}^{(d)} + \mathbf{u}_1^{\mathrm{T}}\mathbf{x}_t) : \|\mathbf{W}_1\|_{1,\infty} \leq \alpha_1, \ \|\mathbf{U}\|_{1,\infty} \leq \beta_1 \right\}, \tag{35}$$

where $\alpha_1, \beta_1$ are nonnegative hyper-parameters.

**The Sista-RNN model** (Wisdom et al., 2017). The hidden state $\mathbf{h}_t^{(l)}$ in Sista-RNN is updated by:

$$\mathbf{h}_t^{(l)} = \begin{cases} \phi\left(\mathbf{W}_1\mathbf{h}_{t-1}^{(d)} + \mathbf{U}_1\mathbf{x}_t\right), & l = 1, \\ \phi\left(\mathbf{W}_2\mathbf{h}_t^{(l-1)} + \mathbf{U}_1\mathbf{x}_t + \mathbf{U}_2\mathbf{h}_{t-1}^{(d)}\right), & l > 1, \end{cases} \tag{36}$$

The real-valued family of functions, $\mathcal{F}_{d,t} : \mathbb{R}^h \times \mathbb{R}^n \mapsto \mathbb{R}$, for the functions $f_{\mathcal{W},\mathbf{U}}^{(d)}$ in layer $d$ is defined by:

$$\mathcal{F}_{d,t} = \left\{ (\mathbf{h}_{t-1}^{(d)}, \mathbf{x}_t) \mapsto \phi\left(\mathbf{w}_2^{\mathrm{T}} f_{\mathcal{W},\mathcal{U}}^{(d-1)}(\mathbf{h}_{t-1}^{(d)}, \mathbf{x}_t) + \mathbf{u}_1^{\mathrm{T}}\mathbf{x}_t + \mathbf{u}_2^{\mathrm{T}}\mathbf{h}_{t-1}^{(d)}\right) \right.$$
$$\left. : \|\mathbf{W}_2\|_{1,\infty} \leq \alpha_2, \|\mathbf{U}\|_{1,\infty} \leq \beta_1, \ \|\mathbf{U}\|_{2,\infty} \leq \beta_2 \right\}, \tag{37}$$

where $\alpha_2, \beta_1, \beta_2$ are nonnegative hyper-parameters. In layer $l = 1$,

$$\mathcal{F}_{1,t} = \left\{ (\mathbf{h}_{t-1}^{(d)}, \mathbf{x}_t) \mapsto \phi\left(\mathbf{w}_1^{\mathrm{T}}\mathbf{h}_{t-1}^{(d)} + \mathbf{u}_1^{\mathrm{T}}\mathbf{x}_t\right) : \|\mathbf{W}_1\|_{1,\infty} \leq \alpha_1, \ \|\mathbf{U}\|_{1,\infty} \leq \beta_1 \right\}, \tag{38}$$

where $\alpha_1, \beta_1$ are nonnegative hyper-parameters.

## D  PROOF OF THEOREM 3.1

*Proof.* We consider the real-valued family of functions $\mathcal{F}_{d,T} : \mathbb{R}^h \times \mathbb{R}^n \mapsto \mathbb{R}$ for the functions $f_{\mathcal{W},\mathbf{U}}^{(d)}$ to update $\mathbf{h}_T^{(d)}$ in layer $d$, time step $T$, defined as

$$\mathcal{F}_{d,T} = \left\{ (\mathbf{h}_{T-1}^{(d)}, \mathbf{x}_T) \mapsto \varPhi(\mathbf{w}_d^{\mathrm{T}} f_{\mathcal{W},\mathcal{U}}^{(d-1)}(\mathbf{h}_{T-1}^{(d)}, \mathbf{x}_T) + \mathbf{u}_d^{\mathrm{T}}\mathbf{x}_T) : \|\mathbf{W}_d\|_{1,\infty} \leq \alpha_d, \ \|\mathbf{U}_d\|_{1,\infty} \leq \beta_d \right\}, \tag{39}$$

where $\mathbf{w}_d, \mathbf{u}_d$ are the corresponding rows from $\mathbf{W}_d, \mathbf{U}_d$, respectively, and $\alpha_l, \beta_l$, with $1 < l \leq d$, are nonnegative hyper-parameters. For the first layer and the first time step, i.e., $l = 1$, $t = 1$, the real-valued family of functions, $\mathcal{F}_{1,1} : \mathbb{R}^h \times \mathbb{R}^n \mapsto \mathbb{R}$, for the functions $f^{(1)}_{\mathcal{W},\mathcal{U}}$ is defined by:

$$\mathcal{F}_{1,1} = \Big\{ (\mathbf{h}_0, \mathbf{x}_1) \mapsto \Phi(\mathbf{w}_1^{\mathrm{T}} \mathbf{h}_0 + \mathbf{u}_1^{\mathrm{T}} \mathbf{x}_1) : \|\mathbf{W}_1\|_{1,\infty} \leq \alpha_1, \ \|\mathbf{U}\|_{1,\infty} \leq \beta_1 \Big\}, \qquad (40)$$

where $\alpha_1, \beta_1$ are nonnegative hyper-parameters. We denote the input layer as $f^{(0)}_{\mathcal{W},\mathcal{U}} = \mathbf{h}_0$ at the first time step. From the definition of Rademacher complexity in (30) and the family of functions in (39) and (40), we obtain:

$$m\mathfrak{R}_S(\mathcal{F}_{d,T}) \qquad (41\mathrm{a})$$

$$\leq \mathbb{E}_{\boldsymbol{\epsilon} \in \{\pm 1\}^m} \left[ \sup_{\substack{\mathcal{W},\mathcal{U} \\ \|\mathbf{w}_d\|_1 \leq \alpha_d \\ \|\mathbf{u}_d\|_1 \leq \beta_d}} \sum_{i=1}^m \epsilon_i \Phi\Big(\mathbf{w}_d^{\mathrm{T}} f^{(d-1)}_{\mathcal{W},\mathcal{U}}(\mathbf{h}_{T-1,i}, \mathbf{x}_{T,i}) + \mathbf{u}_d^{\mathrm{T}} \mathbf{x}_{T,i}\Big) \right]$$

$$\leq \frac{1}{\lambda} \log \exp \left( \mathbb{E}_{\boldsymbol{\epsilon} \in \{\pm 1\}^m} \left[ \sup_{\substack{\mathcal{W},\mathcal{U} \\ \|\mathbf{w}_d\|_1 \leq \alpha_d \\ \|\mathbf{u}_d\|_1 \leq \beta_d}} \lambda \sum_{i=1}^m \epsilon_i \Big(\mathbf{w}_d^{\mathrm{T}} f^{(d-1)}_{\mathcal{W},\mathcal{U}}(\mathbf{h}_{T-1,i}, \mathbf{x}_{T,i}) + \mathbf{u}_d^{\mathrm{T}} \mathbf{x}_{T,i}\Big) \right] \right)$$

$$\leq \frac{1}{\lambda} \log \mathbb{E}_{\boldsymbol{\epsilon} \in \{\pm 1\}^m} \left[ \sup_{\substack{\mathcal{W},\mathcal{U} \\ \|\mathbf{w}_d\|_1 \leq \alpha_d \\ \|\mathbf{u}_d\|_1 \leq \beta_d}} \exp \left( \lambda \sum_{i=1}^m \epsilon_i \Big(\mathbf{w}_d^{\mathrm{T}} f^{(d-1)}_{\mathcal{W},\mathcal{U}}(\mathbf{h}_{T-1,i}, \mathbf{x}_{T,i})\Big) + \lambda \sum_{i=1}^m \epsilon_i \mathbf{u}_d^{\mathrm{T}} \mathbf{x}_{T,i} \right) \right]$$

$$(41\mathrm{b})$$

$$\leq \frac{1}{\lambda} \log \mathbb{E}_{\boldsymbol{\epsilon} \in \{\pm 1\}^m} \left[ \sup_{\substack{\mathcal{W},\mathcal{U} \\ \|\mathbf{w}_d\|_1 \leq \alpha_d}} \exp \left( \lambda \sum_{i=1}^m \epsilon_i \Big(\mathbf{w}_d^{\mathrm{T}} f^{(d-1)}_{\mathcal{W},\mathcal{U}}(\mathbf{h}_{T-1,i}, \mathbf{x}_{T,i})\Big) \right) \right.$$

$$\left. \cdot \sup_{\|\mathbf{u}_d\|_1 \leq \beta_d} \exp \left( \lambda \sum_{i=1}^m \epsilon_i \mathbf{u}_d^{\mathrm{T}} \mathbf{x}_{T,i} \right) \right], \qquad (41\mathrm{c})$$

where $\lambda > 0$ is an arbitrary parameter, Eq. (41b) follows Lemma D.1 for 1-Lipschitz $\Phi$ a long with Inequality (62), and (41c) holds by Inequality (59).

For layer $1 \leq l \leq d$ and time step $t$, let us denote:

$$\Delta^{(l)}_{\mathbf{h}_{t-1},\mathbf{x}_t} = \sup_{\substack{\mathcal{W},\mathcal{U} \\ \|\mathbf{w}_l\|_1 \leq \alpha_l}} \exp \left( \lambda \Lambda_l \sum_{i=1}^m \epsilon_i \Big(\mathbf{w}_l^{\mathrm{T}} f^{(l-1)}_{\mathcal{W},\mathcal{U}}(\mathbf{h}_{t-1,i}, \mathbf{x}_{t,i})\Big) \right), \qquad (42)$$

$$\Delta^{(l)}_{\mathbf{x}_t} = \sup_{\|\mathbf{u}_l\|_1 \leq \beta_l} \exp \left( \lambda \Lambda_l \sum_{i=1}^m \epsilon_i \Big(\mathbf{u}_l^{\mathrm{T}} \mathbf{x}_{t,i}\Big) \right), \qquad (43)$$

where $\Lambda_l$ is defined as follows: $\Lambda_d = 1$, $\Lambda_l = \prod_{k=l+1}^d \alpha_k$ with $1 \leq l \leq d-1$, and $\Lambda_0 = \prod_{k=1}^d \alpha_k$.

Following the Hölder's inequality in (58) in case of $p = 1$ and $q = \infty$ applied to $\mathbf{w}_l^{\mathrm{T}}$ and $f^{(l-1)}_{\mathcal{W},\mathcal{U}}(\mathbf{h}_{t-1,i}, \mathbf{x}_{t,i})$ in (42), respectively, we get:

$$\Delta^{(d)}_{\mathbf{h}_{t-1},\mathbf{x}_t} \qquad (44)$$

$$\leq \sup_{\substack{\mathcal{W},\mathcal{U} \\ \|\mathbf{W}_{d-1}\|_{1,\infty} \leq \alpha_{d-1} \\ \|\mathbf{U}_{d-1}\|_{1,\infty} \leq \beta_{d-1}}} \exp \left( \lambda \alpha_d \left\| \sum_{i=1}^m \epsilon_i \Phi\Big(\mathbf{W}_{d-1} f^{(d-2)}_{\mathcal{W},\mathcal{U}}(\mathbf{h}_{t-1,i}, \mathbf{x}_{t,i}) + \mathbf{U}_{d-1} \mathbf{x}_{t,i}\Big) \right\|_\infty \right)$$

$$\leq \sup_{\substack{\mathcal{W},\mathcal{U} \\ \|\mathbf{w}_{d-1,k}\|_1 \leq \alpha_{d-1} \\ \|\mathbf{u}_{d-1,k}\|_1 \leq \beta_{d-1}}} \exp\left(\lambda\alpha_d \max_{k\in\{1,\cdots,h\}} \left|\sum_{i=1}^{m} \epsilon_i \Phi\left(\mathbf{w}_{d-1,k}^{\mathrm{T}} f_{\mathcal{W},\mathcal{U}}^{(d-2)}(\mathbf{h}_{t-1,i},\mathbf{x}_{t,i}) + \mathbf{u}_{d-1,k}^{\mathrm{T}}\mathbf{x}_{t,i}\right)\right|\right)$$

$$\leq \sup_{\substack{\mathcal{W},\mathcal{U} \\ \|\mathbf{w}_{d-1,k}\|_1 \leq \alpha_{d-1} \\ \|\mathbf{u}_{d-1,k}\|_1 \leq \beta_{d-1}}} \exp\left(\lambda\alpha_d \left|\sum_{i=1}^{m} \epsilon_i \Phi\left(\mathbf{w}_{d-1,k}^{\mathrm{T}} f_{\mathcal{W},\mathcal{U}}^{(d-2)}(\mathbf{h}_{t-1,i},\mathbf{x}_{t,i}) + \mathbf{u}_{d-1,k}^{\mathrm{T}}\mathbf{x}_{t,i}\right)\right|\right). \tag{45}$$

Similarly, from (43), we obtain:

$$\Delta_{\mathbf{x}_t}^{(d)} \leq \sup_{\|\mathbf{u}_d\|_1 \leq \beta_d} \exp\left(\lambda \sum_{i=1}^{m} \epsilon_i \mathbf{u}_d^{\mathrm{T}}\mathbf{x}_{t,i}\right) \leq \exp\left(\lambda\beta_d \left\|\sum_{i=1}^{m} \epsilon_i\mathbf{x}_{t,i}\right\|_\infty\right) \leq \exp\left(\lambda\beta_d \left|\sum_{i=1}^{m} \epsilon_i\mathrm{x}_{\tau,i,\kappa}\right|\right), \tag{46}$$

where $\{\tau,\kappa\} = \underset{t\in\{1,...,T\},j\in\{1,...,n\}}{\mathrm{argmax}} \left|\sum_{i=1}^{m} \epsilon_i\mathrm{x}_{t,i,j}\right|$.

From (41c), (44), and (46), we get:

$$m\mathfrak{R}_S(\mathcal{F}_{d,T})$$

$$\leq \frac{1}{\lambda}\log\left(\underset{\boldsymbol{\epsilon}\in\{\pm1\}^m}{\mathbb{E}}\left[\sup_{\substack{\mathcal{W},\mathcal{U} \\ \|\mathbf{w}_{d-1,k}\|_1 \leq \alpha_{d-1} \\ \|\mathbf{u}_{d-1,k}\|_1 \leq \beta_{d-1}}} \exp\left(\lambda\alpha_d \left|\sum_{i=1}^{m} \epsilon_i \Phi\left(\mathbf{w}_{d-1,k}^{\mathrm{T}} f_{\mathcal{W},\mathcal{U}}^{(d-2)}(\mathbf{h}_{T-1,i},\mathbf{x}_{T,i}) + \mathbf{u}_{d-1,k}^{\mathrm{T}}\mathbf{x}_{T,i}\right)\right|\right.\right.$$

$$\left.\left. + \lambda\beta_d \left|\sum_{i=1}^{m} \epsilon_i\mathrm{x}_{\tau,i,\kappa}\right|\right)\right]\right)$$

$$\leq \frac{1}{\lambda}\log\left(\underset{\boldsymbol{\epsilon}\in\{\pm1\}^m}{\mathbb{E}}\left[\sup_{\substack{\mathcal{W},\mathcal{U} \\ \|\mathbf{w}_{d-1,k}\|_1 \leq \alpha_{d-1} \\ \|\mathbf{u}_{d-1,k}\|_1 \leq \beta_{d-1}}}\left(\right.\right.\right.$$

$$\exp\left(\lambda\alpha_d \sum_{i=1}^{m} \epsilon_i \Phi\left(\mathbf{w}_{d-1,k}^{\mathrm{T}} f_{\mathcal{W},\mathcal{U}}^{(d-2)}(\mathbf{h}_{T-1,i},\mathbf{x}_{T,i}) + \mathbf{u}_{d-1,k}^{\mathrm{T}}\mathbf{x}_{T,i}\right) + \lambda\beta_d \sum_{i=1}^{m} \epsilon_i\mathrm{x}_{\tau,i,\kappa}\right)$$

$$+ \exp\left(\lambda\alpha_d \sum_{i=1}^{m} \epsilon_i \Phi\left(\mathbf{w}_{d-1,k}^{\mathrm{T}} f_{\mathcal{W},\mathcal{U}}^{(d-2)}(\mathbf{h}_{T-1,i},\mathbf{x}_{T,i}) + \mathbf{u}_{d-1,k}^{\mathrm{T}}\mathbf{x}_{T,i}\right) - \lambda\beta_d \sum_{i=1}^{m} \epsilon_i\mathrm{x}_{\tau,i,\kappa}\right)$$

$$+ \exp\left(-\lambda\alpha_d \sum_{i=1}^{m} \epsilon_i \Phi\left(\mathbf{w}_{d-1,k}^{\mathrm{T}} f_{\mathcal{W},\mathcal{U}}^{(d-2)}(\mathbf{h}_{T-1,i},\mathbf{x}_{T,i}) + \mathbf{u}_{d-1,k}^{\mathrm{T}}\mathbf{x}_{T,i}\right) + \lambda\beta_d \sum_{i=1}^{m} \epsilon_i\mathrm{x}_{\tau,i,\kappa}\right)$$

$$\left.\left.\left. + \exp\left(-\lambda\alpha_d \sum_{i=1}^{m} \epsilon_i \Phi\left(\mathbf{w}_{d-1,k}^{\mathrm{T}} f_{\mathcal{W},\mathcal{U}}^{(d-2)}(\mathbf{h}_{T-1,i},\mathbf{x}_{T,i}) + \mathbf{u}_{d-1,k}^{\mathrm{T}}\mathbf{x}_{T,i}\right) - \lambda\beta_d \sum_{i=1}^{m} \epsilon_i\mathrm{x}_{\tau,i,\kappa}\right)\right)\right]\right)$$

$$\leq \frac{1}{\lambda}\log\left(4\underset{\boldsymbol{\epsilon}\in\{\pm1\}^m}{\mathbb{E}}\left[\Delta_{\mathbf{h}_{T-1},\mathbf{x}_T}^{(d-1)}\Delta_{\mathbf{x}_T}^{(d-1)}\exp\left(\beta_d\lambda\sum_{i=1}^{m} \epsilon_i\mathrm{x}_{\tau,i,\kappa}\right)\right]\right) \tag{47a}$$

$$\leq \frac{1}{\lambda}\log\left(4^{d-1}\underset{\boldsymbol{\epsilon}\in\{\pm1\}^m}{\mathbb{E}}\left[\Delta_{\mathbf{h}_{T-1},\mathbf{x}_T}^{(1)}\Delta_{\mathbf{x}_T}^{(1)}\exp\left(\lambda\left(\sum_{l=2}^{d} \beta_l\Lambda_l\right)\sum_{i=1}^{m} \epsilon_i\mathrm{x}_{\tau,i,\kappa}\right)\right]\right) \tag{47b}$$

$$\leq \frac{1}{\lambda}\log\left(4^{d-1}\underset{\boldsymbol{\epsilon}\in\{\pm1\}^m}{\mathbb{E}}\left[\exp\left(\lambda\left(\sum_{l=2}^{d} \beta_l\Lambda_l\right)\sum_{i=1}^{m} \epsilon_i\mathrm{x}_{\tau,i,\kappa}\right)\sup_{\|\mathbf{w}_1\|_1 \leq \alpha_1}\exp\left(\lambda\Lambda_1\sum_{i=1}^{m} \epsilon_i\left(\mathbf{w}_1^{\mathrm{T}}\mathbf{h}_{T-1,i}\right)\right)\right.\right.$$

$$\left.\left. \cdot \sup_{\|\mathbf{u}_1\|_1 \leq \beta_1}\exp\left(\lambda\Lambda_1\sum_{i=1}^{m} \epsilon_i\left(\mathbf{u}_1^{\mathrm{T}}\mathbf{x}_{T,i}\right)\right)\right]\right) \tag{47c}$$

$$\leq \frac{1}{\lambda} \log \left( 4^{d-1} \mathop{\mathbb{E}}_{\boldsymbol{\epsilon} \in \{\pm 1\}^m} \left[ \exp \left( \lambda \Big( \sum_{l=2}^{d} \beta_l \Lambda_l \Big) \sum_{i=1}^{m} \epsilon_i \mathrm{x}_{\tau,i,\kappa} \right) \sup_{\substack{\mathcal{W},\mathcal{U} \\ \|\mathbf{w}_d\|_1 \leq \alpha_d \\ \|\mathbf{u}_d\|_1 \leq \beta_d}} \exp \left( \lambda \Lambda_0 \Big\| \sum_{i=1}^{m} \epsilon_i \mathbf{h}_{T-1,i} \Big\|_{\infty} \right) \right. \right.$$

$$\left. \left. \cdot \exp \left( \lambda \beta_1 \Lambda_1 \Big\| \sum_{i=1}^{m} \epsilon_i \mathbf{x}_{T,i} \Big\|_{\infty} \right) \right] \right) \tag{47d}$$

$$\leq \frac{1}{\lambda} \log \left( 4^{d} \mathop{\mathbb{E}}_{\boldsymbol{\epsilon} \in \{\pm 1\}^m} \left[ \exp \left( \lambda \Big( \sum_{l=1}^{d} \beta_l \Lambda_l \Big) \sum_{i=1}^{m} \epsilon_i \mathrm{x}_{\tau,i,\kappa} \right) \right. \right.$$

$$\left. \left. \cdot \sup_{\substack{\mathcal{W},\mathcal{U} \\ \|\mathbf{w}_d\|_1 \leq \alpha_d \\ \|\mathbf{u}_d\|_1 \leq \beta_d}} \exp \left( \lambda \Lambda_0 \sum_{i=1}^{m} \epsilon_i \Phi \Big( \mathbf{w}_d^{\mathrm{T}} f_{\mathcal{W},\mathcal{U}}^{(d-1)}(\mathbf{h}_{T-2,i}, \mathbf{x}_{T-1,i}) + \mathbf{u}_d^{\mathrm{T}} \mathbf{x}_{T-1,i} \Big) \right) \right] \right), \tag{47e}$$

where (47a) holds by inequality (59), and (47b) follows by repeating the process from layer $d-1$ to layer 1 for time step $T$. Furthermore, (47c) is obtained as the beginning of the process for time step $T-1$ and (47d) follows inequality (58).

Proceeding by repeating the above procedure in (47e) from time step $T-1$ to time step 1, we get:
$$m \mathfrak{R}_S(\mathcal{F}_{d,T})$$

$$\leq \frac{1}{\lambda} \log \left( 4^{dT} \mathop{\mathbb{E}}_{\boldsymbol{\epsilon} \in \{\pm 1\}^m} \left[ \exp \left( \lambda \Big( \sum_{l=1}^{d} \beta_l \Lambda_l \Big) \Big( \frac{\Lambda_0^T - 1}{\Lambda_0 - 1} \Big) \sum_{i=1}^{m} \epsilon_i \mathrm{x}_{\tau,i,\kappa} \right) \exp \left( \lambda \Lambda_0^T \Big\| \sum_{i=1}^{m} \epsilon_i \mathbf{h}_0 \Big\|_{\infty} \right) \right] \right). \tag{48}$$

Let us denote $\mu = \operatorname*{argmax}_{j \in \{1,\ldots,h\}} \Big| \sum_{i=1}^{m} \epsilon_i \mathrm{h}_{0,j} \Big|$, from (48), we have:

$$m \mathfrak{R}_S(\mathcal{F}_{d,T})$$

$$\leq \frac{1}{\lambda} \log \left( 4^{dT} \mathop{\mathbb{E}}_{\boldsymbol{\epsilon} \in \{\pm 1\}^m} \left[ \exp \left( \lambda \Big( \sum_{l=1}^{d} \beta_l \Lambda_l \Big) \Big( \frac{\Lambda_0^T - 1}{\Lambda_0 - 1} \Big) \sum_{i=1}^{m} \epsilon_i \mathrm{x}_{\tau,i,\kappa} \right) \exp \left( \lambda \Lambda_0^T \sum_{i=1}^{m} \epsilon_i \mathrm{h}_{0,\mu} \right) \right] \right)$$

$$\leq \frac{2dT \log 2}{\lambda} + \frac{1}{2\lambda} \log \left( \mathop{\mathbb{E}}_{\boldsymbol{\epsilon} \in \{\pm 1\}^m} \left[ \exp \left( \lambda \Big( \sum_{l=1}^{d} \beta_l \Lambda_l \Big) \Big( \frac{\Lambda_0^T - 1}{\Lambda_0 - 1} \Big) \sum_{i=1}^{m} \epsilon_i \mathrm{x}_{\tau,i,\kappa} \right) \right. \right.$$

$$\left. \left. \cdot \exp \left( \lambda \Lambda_0^T \sum_{i=1}^{m} \epsilon_i \mathrm{h}_{0,\mu} \right) \right]^2 \right)$$

$$\leq \frac{2dT \log 2}{\lambda} + \frac{1}{2\lambda} \log \mathop{\mathbb{E}}_{\boldsymbol{\epsilon} \in \{\pm 1\}^m} \left[ \exp \left( 2\lambda \Big( \sum_{l=1}^{d} \beta_l \Lambda_l \Big) \Big( \frac{\Lambda_0^T - 1}{\Lambda_0 - 1} \Big) \sum_{i=1}^{m} \epsilon_i \mathrm{x}_{\tau,i,\kappa} \right) \right]$$

$$+ \frac{1}{2\lambda} \log \mathop{\mathbb{E}}_{\boldsymbol{\epsilon} \in \{\pm 1\}^m} \left[ \exp \left( 2\lambda \Lambda_0^T \sum_{i=1}^{m} \epsilon_i \mathrm{h}_{0,\mu} \right) \right] \tag{49a}$$

$$\leq \frac{2dT \log 2}{\lambda} + \frac{1}{2\lambda} \log \sum_{j=1}^{n} \mathop{\mathbb{E}}_{\boldsymbol{\epsilon} \in \{\pm 1\}^m} \left[ \exp \left( 2\lambda \Big( \sum_{l=1}^{d} \beta_l \Lambda_l \Big) \Big( \frac{\Lambda_0^T - 1}{\Lambda_0 - 1} \Big) \sum_{i=1}^{m} \epsilon_i \mathrm{x}_{\tau,i,j} \right) \right]$$

$$+ \frac{1}{2\lambda} \log \sum_{j=1}^{h} \mathop{\mathbb{E}}_{\boldsymbol{\epsilon} \in \{\pm 1\}^m} \left[ \exp \left( 2\lambda \Lambda_0^T \sum_{i=1}^{m} \epsilon_i \mathrm{h}_{0,j} \right) \right] \tag{49b}$$

$$\leq \frac{2dT \log 2}{\lambda} + \frac{1}{2\lambda} \log \sum_{j=1}^{n} \prod_{i=1}^{m} \mathop{\mathbb{E}}_{\boldsymbol{\epsilon} \in \{\pm 1\}^m} \left[ \exp \left( 2\lambda \Big( \sum_{l=1}^{d} \beta_l \Lambda_l \Big) \Big( \frac{\Lambda_0^T - 1}{\Lambda_0 - 1} \Big) \epsilon_i \mathrm{x}_{\tau,i,j} \right) \right]$$

$$+ \frac{1}{2\lambda} \log \sum_{j=1}^{h} \prod_{i=1}^{m} \mathop{\mathbb{E}}_{\boldsymbol{\epsilon} \in \{\pm 1\}^m} \left[ \exp \left( 2\lambda \Lambda_0^T \epsilon_i \mathrm{h}_{0,j} \right) \right]$$

$$
\begin{aligned}
\leq\; & \frac{2dT\log 2}{\lambda} + \frac{1}{2\lambda}\log\sum_{j=1}^{n}\prod_{i=1}^{m}\left[\frac{1}{2}\exp\left(2\lambda\Big(\sum_{l=1}^{d}\beta_l\varLambda_l\Big)\Big(\frac{\varLambda_0^T-1}{\varLambda_0-1}\Big)\mathrm{x}_{\tau,i,j}\right)\right.\\
& \left. + \frac{1}{2}\exp\left(-2\lambda\Big(\sum_{l=1}^{d}\beta_l\varLambda_l\Big)\Big(\frac{\varLambda_0^T-1}{\varLambda_0-1}\Big)\mathrm{x}_{\tau,i,j}\right)\right]\\
& + \frac{1}{2\lambda}\log\sum_{j=1}^{h}\prod_{i=1}^{m}\left[\frac{1}{2}\exp\left(2\lambda\varLambda_0^T\mathrm{h}_{0,j}\right) + \frac{1}{2}\exp\left(-2\lambda\varLambda_0^T\mathrm{h}_{0,j}\right)\right]\\
\leq\; & \frac{2dT\log 2}{\lambda} + \frac{1}{2\lambda}\log\sum_{j=1}^{n}\left[\exp\left(2\lambda^2\Big(\sum_{l=1}^{d}\beta_l\varLambda_l\Big)^2\Big(\frac{\varLambda_0^T-1}{\varLambda_0-1}\Big)^2\sum_{i=1}^{m}x_{\tau,i,j}^2\right)\right]\\
& + \frac{1}{2\lambda}\log\sum_{j=1}^{h}\left[\exp\left(2\lambda^2\varLambda_0^{2T}\sum_{i=1}^{m}h_{0,j}^2\right)\right] \tag{49c}\\
\leq\; & \frac{2dT\log 2}{\lambda} + \frac{\log n}{2\lambda} + \lambda\Big(\sum_{l=1}^{d}\beta_l\varLambda_l\Big)^2\Big(\frac{\varLambda_0^T-1}{\varLambda_0-1}\Big)^2 mB_{\mathbf{x}}^2 + \frac{\log h}{2\lambda} + \lambda\varLambda_0^{2T}m\|\mathbf{h}_0\|_\infty^2\\
\leq\; & \frac{2dT\log 2 + \log\sqrt{n} + \log\sqrt{h}}{\lambda} + \lambda\left(\Big(\sum_{l=1}^{d}\beta_l\varLambda_l\Big)^2\Big(\frac{\varLambda_0^T-1}{\varLambda_0-1}\Big)^2 mB_{\mathbf{x}}^2 + \varLambda_0^{2T}m\|\mathbf{h}_0\|_\infty^2\right), \tag{49d}
\end{aligned}
$$

where (49a) follows inequality (61), and (49b) holds by replacing with $\sum_{j=1}^{n}$ and $\sum_{j=1}^{h}$, respectively. In addition, (49c) follows (60), and (49d) is obtained by the following definition: At time step $t$, we define $\mathbf{X}_t \in \mathbb{R}^{n\times m}$, a matrix composed of $m$ columns from the $m$ input vectors $\{\mathbf{x}_{t,i}\}_{i=1}^{m}$; we also define $\|\mathbf{X}_t\|_{2,\infty} = \sqrt{\max_{k\in\{1,\dots,n\}}\sum_{i=1}^{m}\mathrm{x}_{t,i,k}^2} \leq \sqrt{m}B_{\mathbf{x}}$, representing the maximum of the $\ell_2$-norms of the rows of matrix $\mathbf{X}_t$, and $\|\mathbf{h}_0\|_\infty = \max_j |\mathrm{h}_{0,j}|$.

Choosing $\lambda = \sqrt{\dfrac{2dT\log 2 + \log\sqrt{n} + \log\sqrt{h}}{\Big(\sum_{l=1}^{d}\beta_l\varLambda_l\Big)^2\Big(\frac{\varLambda_0^T-1}{\varLambda_0-1}\Big)^2 mB_{\mathbf{x}}^2 + \varLambda_0^{2T}m\|\mathbf{h}_0\|_\infty^2}}$, we achieve the upper bound:

$$
\mathfrak{R}_S(\mathcal{F}_{d,T}) \leq \sqrt{\frac{2(4dT\log 2 + \log n + \log h)}{m}\left(\Big(\sum_{l=1}^{d}\beta_l\varLambda_l\Big)^2\Big(\frac{\varLambda_0^T-1}{\varLambda_0-1}\Big)^2 B_{\mathbf{x}}^2 + \varLambda_0^{2T}\|\mathbf{h}_0\|_\infty^2\right)}. \tag{50}
$$

It can be noted that $\mathfrak{R}_S(\mathcal{F}_{d,T})$ in (50) is derived for the real-valued functions $\mathcal{F}_{d,T}$. For the vector-valued functions $\mathcal{F}_{d,T} : \mathbb{R}^h \times \mathbb{R}^n \mapsto \mathbb{R}^h$ (in Theorem 3.1), we apply the contraction lemma (Lemma D.1) to a Lipschitz loss to obtain the complexity of such vector-valued functions by means of the complexity of the real-valued functions. Specifically, in Theorem 3.1, under the assumption of the 1-Lipschitz loss function and from Theorem C.1, Lemma D.1, we complete the proof.

$\square$

## D.1 Comparison with existing generalization bounds

Recent works have established generalization bounds for RNN models with a single recurrent layer ($d = 1$) using Rademacher complexity [see FastRNN in Kusupati et al. (2018)] or PAC-Bayes theory [see SpectralRNN in Zhang et al. (2018)]. We re-state these generalization bounds below and apply Theorem 3.1 with $d = 1$ to compare with our bound for reweighted-RNN.

**FastRNN** (Kusupati et al., 2018). The hidden state $\mathbf{h}_t$ of FastRNN is updated as follows:

$$
\begin{aligned}
\tilde{\mathbf{h}}_t &= \phi(\mathbf{W}\mathbf{h}_{t-1} + \mathbf{U}\mathbf{x}_t)\\
\mathbf{h}_t &= a\tilde{\mathbf{h}}_t + b\mathbf{h}_{t-1}, \tag{51}
\end{aligned}
$$

where $0 \leq a, b \leq 1$ are trainable parameters parameterized by the sigmoid function. Under the assumption that $a + b = 1$, the Rademacher complexity $\Re_S(\mathcal{F}_T)$ of the class $\mathcal{F}_T$ of FastRNN (Kusupati et al., 2018), with $\|\mathbf{W}\|_F \leq \alpha_F$, $\|\mathbf{U}\|_F \leq \beta_F$, and $\|\mathbf{x}_t\|_2 \leq B$, is given by

$$\Re_S(\mathcal{F}_T) \leq \frac{2a}{\sqrt{m}} B \beta_F \left( \frac{(1 + a(2\alpha_F - 1))^{T+1} - 1}{a(2\alpha_F - 1)} \right). \tag{52}$$

Alternatively, under the additional assumption that $a \leq \frac{1}{2(2\alpha_F - 1)T}$, the bound in Kusupati et al. (2018) becomes:

$$\Re_S(\mathcal{F}_T) \leq \frac{2a}{\sqrt{m}} B \beta_F \left( \frac{2a(2\alpha_F - 1)(T + 1) - 1}{(2\alpha_F - 1)} \right). \tag{53}$$

**SpectralRNN** (Zhang et al., 2018). The hidden state $\mathbf{h}_t$ and output $\mathbf{y}_t \in \mathbb{R}^{n_\mathbf{y}}$ of SpectralRNN are computed as:

$$\begin{aligned} \mathbf{h}_t &= \phi(\mathbf{W}\mathbf{h}_{t-1} + \mathbf{U}\mathbf{x}_t) \\ \mathbf{y}_t &= \mathbf{Y}\mathbf{h}_t, \end{aligned} \tag{54}$$

where $\mathbf{Y} \in \mathbb{R}^{n_\mathbf{y} \times h}$. The generalization error in Zhang et al. (2018) is derived for a classification problem. For any $\delta > 0, \gamma > 0$, with probability $\geq 1 - \delta$ over a training set $S$ of size $m$, the generalization error (Zhang et al., 2018) of SpectralRNN is bounded by

$$\mathcal{O}\left( \sqrt{ \frac{ \frac{B^2 T^4 \xi \ln(\xi)}{\gamma^2} (\|\mathbf{W}\|_F^2 + \|\mathbf{U}\|_F^2 + \|\mathbf{Y}\|_F^2) \cdot \zeta + \ln \frac{m}{\delta} }{m} } \right), \tag{55}$$

where $\zeta = \max\{\|\mathbf{W}\|_2^{2T-2}, 1\} \max\{\|\mathbf{U}\|_2^2, 1\} \max\{\|\mathbf{Y}\|_2^2, 1\}$ and $\xi = \max\{n, n_\mathbf{y}, h\}$.

**Reweighted-RNN**. Based on Theorem 3.1, under the assumption that the initial hidden state $\mathbf{h}_0 = \mathbf{0}$, the Rademacher complexity of reweighted-RNN with $d = 1$ is bounded as

$$\Re_S(\mathcal{F}_{1,T}) \leq \sqrt{ \frac{4T \log 2 + \log n + \log h}{m} } \left( \sqrt{2} \beta_1 \frac{\alpha_1^T - 1}{\alpha_1 - 1} B_\mathbf{x} \right). \tag{56}$$

We observe that the bound of SpectralRNN in (55) depends on $T^2$, whereas the bound of FastRNN either grows exponentially with $T$ (52) or is proportional to $T$ (53). Our bound (56) depends on $\sqrt{T}$, given that the second factor in (56) is only dependent on the norm constraints $\alpha_1$, $\beta_1$ and the input training data; meaning that it is tighter than those of SpectralRNN and FastRNN in terms of the number of time steps.

### D.2 BACKGROUND ON RADEMACHER COMPLEXITY CALCULUS

The contraction lemma in Shalev-Shwartz & Ben-David (2014) gives the Rademacher complexity of the composition of a class of functions with $\rho$-Lipschitz functions.

**Lemma D.1.** *(Shalev-Shwartz & Ben-David, 2014, Lemma 26.9—Contraction lemma)*
*Let $\mathcal{F}$ be a set of functions, $\mathcal{F} = \{f : \mathcal{X} \mapsto \mathbb{R}\}$, and $\Phi_1, ..., \Phi_m$, $\rho$-Lipschitz functions, namely, $|\Phi_i(\alpha) - \Phi_i(\beta)| \leq \rho|\alpha - \beta|$ for all $\alpha, \beta \in \mathbb{R}$ for some $\rho > 0$. For any sample set $S$ of $m$ points $\mathbf{x}_1, ..., \mathbf{x}_m \in \mathcal{X}$, let $(\Phi \circ f)(\mathbf{x}_i) = \Phi(f(\mathbf{x}_i))$. Then,*

$$\frac{1}{m} \mathop{\mathbb{E}}_{\boldsymbol{\epsilon} \in \{\pm 1\}^m} \left[ \sup_{f \in \mathcal{F}} \sum_{i=1}^m \epsilon_i (\Phi \circ f)(\mathbf{x}_i) \right] \leq \frac{\rho}{m} \mathop{\mathbb{E}}_{\boldsymbol{\epsilon} \in \{\pm 1\}^m} \left[ \sup_{f \in \mathcal{F}} \sum_{i=1}^m \epsilon_i f(\mathbf{x}_i) \right], \tag{57}$$

*alternatively, $\Re_S(\boldsymbol{\Phi} \circ \mathcal{F}) \leq \rho \Re_S(\mathcal{F})$, where $\boldsymbol{\Phi}$ denotes $\Phi_1(\mathbf{x}_1), ..., \Phi_m(\mathbf{x}_m)$ for $S$.*

**Proposition D.2.** *(Mohri et al., 2018, Proposition A.1—Hölder's inequality)*
*Let $p, q \geq 1$ be conjugate: $\frac{1}{p} + \frac{1}{q} = 1$. Then, for all $\mathbf{x}, \mathbf{y} \in \mathbb{R}^n$,*

$$\|\mathbf{x} \cdot \mathbf{y}\|_1 \leq \|\mathbf{x}\|_p \|\mathbf{y}\|_q, \tag{58}$$

*with the equality when $|\mathrm{y}_i| = |\mathrm{x}_i|^{p-1}$ for all $i \in [1, n]$.*

**Supporting inequalities**:

(i) If A, B are sets of positive real numbers, then:
$$\sup(AB) = \sup(A) \cdot \sup(B). \tag{59}$$

(ii) Given $x \in \mathbb{R}$, we have:
$$\frac{\exp(x) + \exp(-x)}{2} \leq \exp(x^2/2). \tag{60}$$

(iii) Let X and Y be random variables, the Cauchy–Bunyakovsky–Schwarz inequality gives:
$$(\mathbb{E}[XY])^2 \leq \mathbb{E}[X^2] \cdot \mathbb{E}[Y^2]. \tag{61}$$

(iv) If $\psi$ is a convex function, the Jensen's inequality gives:
$$\psi(\mathbb{E}[X]) \leq \mathbb{E}[\psi(X)]. \tag{62}$$

# E    ADDITIONAL EXPERIMENTS

We test our model on three popular tasks for RNNs, namely the sequential pixel MNIST classification, the adding task, and the copy task (Le et al., 2015; Arjovsky et al., 2016; Zhang et al., 2018).

**Sequential pixel MNIST and permuted pixel MNIST classification**. This task aims to classify MNIST images to a class label. MNIST images are formed by a $28 \times 28$ gray-scale image with a label from 0 to 9. We use the reweighted-RNN along with a softmax for category classification. We set $d = 5$ layers and $h = 256$ hidden units for the reweighted-RNN. We consider two scenarios: the first one where the pixels of each MNIST image are read in the order from left-to-right and bottom-to-top and the second one where the pixels of each MNIST image are randomly permuted. The classification accuracy results are shown in Fig. 7(a) (for pixel MNIST) and Fig. 7(b) (for permuted pixel MNIST).

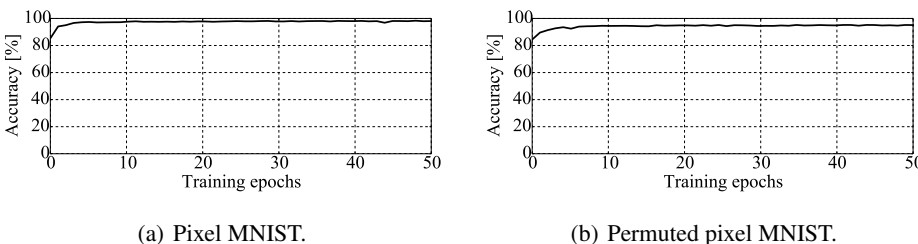

(a) Pixel MNIST.                    (b) Permuted pixel MNIST.

Figure 7: Reweighted-RNN on the (a) pixel-MNIST classification and (b) permuted pixel MNIST classification: Average classification accuracy vs. training epoches on the validation set.

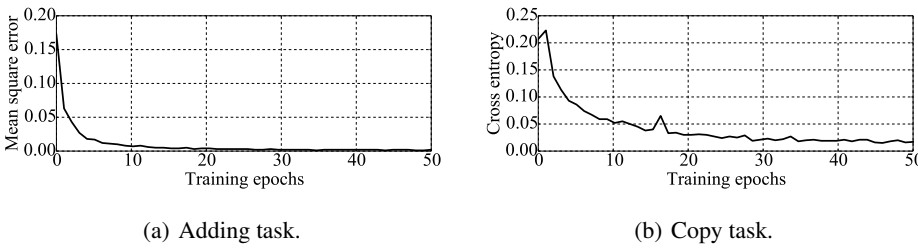

(a) Adding task.                    (b) Copy task.

Figure 8: Reweighted-RNN on the (a) adding task with average mean square error and the (b) copy task with average cross entropy vs. training epoches on the validation set.

**Adding Task**. The task inputs two sequences of length $T$. The first sequence consists of entries that are uniformly sampled from $[0, 1]$. The second sequence comprises two entries of 1 and the

remaining entries of 0, in which the first entry of 1 is randomly located in the first half of the sequence and the second entry of 1 is randomly located in the second half. The output is the sum of the two entrie of the first sequence, where are located in the same posisions of the entries of 1 in the second sequence. We also use the reweighted-RNN with $d = 5$ layers and $h = 256$ hidden units for the input sequences of length $T = 300$. Fig. 8(a) shows the mean square error versus training epoches on the validation set.

**Copy task**. We consider an input sequence $\mathbf{x} \in \mathbb{A}^{T+20}$ (Zhang et al., 2018), where $\mathbb{A} = \{a_0, \cdots, a_9\}$. $x_0, \cdots, x_9$ are uniformly sampled from $\{a_0, \cdots, a_7\}$, $x_{T+10} = a_9$, and the remaining $x_i$ are set to $a_8$. The purpose of this task is to copy $x_0, \cdots, x_9$ to the end of the output sequence $\mathbf{y} \in \mathbb{A}^{T+20}$ given a time lag $T$, i.e., $\{y_{T+10}, \cdots, y_{T+19}\} \equiv \{x_0, \cdots, x_9\}$ and the remaining $y_i$ are equal to $a_8$. We set the reweighted-RNN $d = 5$ layers and $h = 256$ hidden units for the input sequences of a time lag $T = 100$. Fig. 8(b) shows the cross entropy versus training epoches on the validation set.

