# OpenReview forum: "A Deep Recurrent Neural Network via Unfolding Reweighted l1-l1 Minimization"
_ICLR.cc/2020/Conference — Reject_

### Official Review · AnonReviewer1 · 2019-10-23
**Official Blind Review #1**

**Rating:** 3

**Review:**


Strength:
This paper proposes a new reweighted-RNN by unfolding a reweighted L1-L1 minimization problem. It develops an iterative algorithm to solve the reweighted L1-L1 minimization problem, where the soft-thresholding functions can be adaptively learned. This paper provides the generalization error bound for deep RNNs and shows that the proposed reweighted-RNN has a lower generalization error bound. In addition, the paper shows that the proposed algorithm can be applied to video-frame reconstruction and achieves favorable results against state-of-the-art methods. The paper is well organized, and the motivation is clear.

Weakness:
The effectiveness of the reweighted L1-L1 minimization method should be better explained and evaluated. It is not clear why the reweighted L1-L1 regularization is better than the L1-L1 regularization. In addition, the experimental evaluation does not support this claim well. The authors should compare the baseline method which uses the  L1-L1 regularization in their framework instead of directly comparing the proposed algorithm with [Le et al., 2019] as there exist differences in the algorithm design. This is an important baseline.

As claimed by the authors, the proposed reweighted-RNN has different sets of {W_l;U_l} for each hidden layer. This will definitively increase the model size when the depth increases. The authors should clarify whether the performance gains due to the only use of large model parameters.

Overall, this paper proposes an effective reweighted-RNN model based on the solver of a reweighted L1-L1 minimization. Theoretical analysis and experimental results are provided. I would be willing to increase the score if these problems are solved in the authors’ response.


**Experience Assessment:**

I have published one or two papers in this area.

**Review Assessment: Checking Correctness Of Derivations And Theory:**

I assessed the sensibility of the derivations and theory.

**Review Assessment: Checking Correctness Of Experiments:**

I assessed the sensibility of the experiments.

**Review Assessment: Thoroughness In Paper Reading:**

I read the paper at least twice and used my best judgement in assessing the paper.

---

> ### Author Response · Authors · 2019-11-15
> **Response to Reviewer #1**
>
> Thank you very much for your review and for recognizing the strengths of our work. Below, we would like to address your concerns:
>
> -	Regarding why reweighted l1 regularization is better than l1 regularization, we refer to the explanations by Candès et al. (2008) and Luong et al. (2018). The l1 minimization is a relaxation of the L0 minimization problem for recovering a sparse signal. The l1 norm depends on the magnitude of the nonzero signal coefficients, while the L0 norm only counts the number of nonzero coefficients in the signal. Hence, as proposed by Candès et al. (2008), the weights in the reweighted version of l1 minimization are designed to reduce the impact of the magnitude of the nonzero elements, thereby leading to a solution that approximates better the one obtained with L0 minimization.
>
> -	It is not clear to us what design differences exist between the l1-l1 RNN [Le et al., 2019] and a deep-unfolded l1-l1 network. We believe that if we omit the additional reweighted terms Z and g in our minimization problem (3), the problem boils down to l1-l1 minimization and the model resulting by applying deep unfolding will be the same as the l1-l1 RNN [Le et al., 2019]. We mention this in Section 2, in the text after Algorithm 1.
>
> -	We agree with your statement that our model size will increase when increasing the depth. However, the performance gain of the proposed model over the l1-l1-RNN model is not because of just adding extra model parameters in the latter. We start from the fundamental idea that by applying reweighting (Candès et al., 2008) the solution of Problem (3) is a more accurate sparse representation compared to the solution of the l1-l1 minimization problem in Le et al. (2019). The extra parameters in the Reweighted-RNN model are the result of performing deep unfolding of the reweighted minimization algorithm. The Reweighted-RNN model introduces the following innovations compared to the l1-l1-RNN model. Firstly, the Reweighted-RNN uses a different Z_l matrix per iteration/layer (l=1,...,d) to re-parameterize the update terms (see Step 6 in Algorithm 1). Due to Z_l (l=1,...,d), Reweighted-RNN has a different weight coupling per layer l compared to the l1-l1-RNN. Secondly, because of the learned set of vectors g_l (l=1,...,d), Reweighted-RNN applies a different proximal operator to each element u per iteration l. This translates to that the Reweighted-RNN applies a different nonlinearity to each activation in each layer, the form of which is learned from data. This is fundamentally different from the l1-l1-RNN model which applies the same nonlinear function to all activations in all layers. This is clarified in Section 2, in the text after Algorithm 1. Last but not least, the over-parameterization of Reweighted-RNN is supported by theory. The derived generalization error bounds (the first such bounds for deep RNN models and for deep RNNs designed by unfolding) show that the over-parameterization of the Reweighted-RNN helps to improve performance compared to other RNNs including deep unfolding ones.

---

### Official Review · AnonReviewer5 · 2019-11-01
**Official Blind Review #5**

**Rating:** 6

**Review:**

Authors proposed a deep RNN via unfolding reweighted l1-l1 minimization, where reweighted l1-l1 minimization algorithms are applied to a video task.
Overall, the paper is well explained in a theoretical part and exhibits a good result compared with other conventional RNN methods in the experiment. In Section 3, authors formulate Rademacher complexities for both conventional and proposed method, which shows the generalization performance of the proposed method when d increases. And this is empirically highlighted in Table 3 in Section 4.

Major points:
Section 1:
-	First part of the introduction can be confusing because Eq. (1) sounds like representing dictionary learning framework (plus DNN is immediately described after Eq. (1) instead of RNN) and RNN is not explicitly written. It should be clearly written and flow should be considered.
Section 2:
-	It is hard to get how parameter g in Eq. (3) derives.
Section 3:
-	How to build network depth d for the network? A figure should be required.
Section 4:
-	Even though previous papers (e.g., Wisdom et al. and Le et al.) just focus on single dataset like moving MNIST, I believe testing on language data is also quite important (this is a full paper and exhaustive experiments should be mandatory). For example, it may be good to use Penn TreeBank dataset to make a comparison.
-	In Table 3, how did you set LSTM deeper? Is it a stacked LSTM?
-	Existing RNN methods should include other variations of LSTM (in particular, SOTA methods are welcomed) such as bidirectional LSTM and LSTM with attention mechanism. It should be better to compare with these methods.

Appendix:
-	It would be helpful for readers to show interpretabilities of the model additionally. For example, visualizing features from each RNN model would be beneficial.

Minor points:
-	After introduction of unfolding reweighted l1-l1 minimization, how did the computational cost increase compared to previous l1-l1 minimization?
-	In Section3, for easiness to readers, it may be good to briefly summarize how does the predictor’s generalizability and Rademacher complexities relate.

**Experience Assessment:**

I do not know much about this area.

**Review Assessment: Checking Correctness Of Derivations And Theory:**

I did not assess the derivations or theory.

**Review Assessment: Checking Correctness Of Experiments:**

I assessed the sensibility of the experiments.

**Review Assessment: Thoroughness In Paper Reading:**

I read the paper at least twice and used my best judgement in assessing the paper.

---

> ### Author Response · Authors · 2019-11-15
> **Response to Reviewer #5**
>
> Thank you for the comments and suggestions on the manuscript. Please find below our responses to your points:
>
> Major points:
> Section 1: Following this comment, we have revised the introduction and now mention the RNN model in the second paragraph. In addition, we have explicitly added Eq. (3) characterizing the considered RNN.
>
> Section 2: The vector g in Eq. (3) is our proposed reweighted parameter. The motivation is that by applying reweighting (Candès et al., 2008), the solution of Problem (3) is a more accurate sparse representation compared to the solution of the l1-l1 minimization problem in Le et al. (2019). After unfolding the reweighted minimization algorithms, g is also a trainable parameter in our RNN.
>
> Section 3: If we understand your comment correctly, you would like to know how we determined the network depth d for our architecture. d corresponds to the number of iterations in Algorithm 1, and we did not set a value for d but rather experimentally assessed our network with different network depths d (we refer to the experiments reported in Table 3). In case your comment refers to a better illustration of the proposed architecture, we have updated Figure 2 accordingly so as to show how the depth d of our network is developed.
>
> Section 4:
> -       We agree that language understanding tasks are important applications of RNNs and should be considered when benchmarking a new RNN architecture. However, our minimization algorithm (which solves the reweighted l1-l1 minimization problem and which yields the proposed RNN model by deep unfolding) is formalized based on leveraging the specific structures present in video data (namely, the first l1 term for the sparsity in frame representation and the second term for the correlation of consecutive frame representations). Therefore, our model is better suited to applications in video. This does not mean that unfolded RNNs are not applicable to other types of data, but in such applications one would need to revise the minimization objective so as to accurately express the data structure. Motivated by this comment (and Question 3 of Reviewer 1), we report additional experiments of our RNN model in popular RNN tasks, namely, the pixel MNIST classification task, the adding task, and the copy task. We refer to Appendix E for further details.
>
> -	Indeed, in Table 3 we consider a stacked LSTM. Specifically, we stack all models (including LSTM) except unfolding-based ones in the same way as a stacked-RNN (i.e. replacing the vanilla RNN cell with the corresponding cell). Regarding the deep unfolding-based models, the underlying minimization algorithms determine the connections between the layers.
>
> -	Thank you for this comment. Adding bi-directional connections or attention mechanisms would indeed increase the effectiveness of an LSTM. However, we would argue that these additions could be applied to any RNN architecture (not only LSTM) to obtain better performance. In our experiments, we want to limit additional components in the benchmarked models (except those that stabilize training) so that we allow for a fair comparison between different RNN architectures.
>
> Appendix:
> We thank the reviewer for this suggestion. We will plan to visualize these features in our subsequent work and also to focus further on the explainability aspects of the architecture.
>
> Minor points:
> -	In our experiments, the training time for Reweighted RNN with the default settings is 3,521 seconds, an increase in terms of time complexity compared to the baseline of l1-l1-RNN (Le et al. 2019) with 2,985 seconds. We wish to also refer to our answer to the 1st comment of Reviewer 1.
>
> -	Following this comment, we have clarified this aspect after Theorem 3.1 in Section 3, where the text reads as follows “The generalization error in Eq. (14) is bounded by the Rademacher complexity, which depends on the training set S. If the Rademacher complexity is small, the network can be learned with a small generalization error.”

---

### Official Review · AnonReviewer4 · 2019-11-03
**Official Blind Review #4**

**Rating:** 8

**Review:**

This paper proposes a novel method to solve the sequential signal reconstruction problem. The method is based on the deep unfolding methods and incorporates the reweighting mechanism. Additionally, they derive the generalization error bound and show how their over-parameterized reweighting RNNs ensure good generalization. Lastly, the experiments on the task of video sequence reconstruction suggest the superior performance of the proposed method.

I recommend the paper to be accepted for mainly two reasons. First, they derive a tighter generalization bound for deep RNNs; Second, the experiment results align with the theory and show the continuous improvements when increasing the depth of RNNs.

Questions:
1. How is the computation complexity of the proposed method when compared with other methods? Will the reweighting l1-l1 norm significantly increase the computation time?
2. The experiments show that increasing the depth and/or width of the networks yields better performance, however, is there a boundary for such performance gain? For example, if the depth continues increasing, will the proposed method suffer the similar problem as other methods (performance does not improve or even degrade)?
3. As the MOVING MNIST dataset is from a relatively simple and special domain, is it possible to reproduce the similar performance gain on other more realistic datasets?
4. Are there any known limitations of the proposed method?

**Experience Assessment:**

I do not know much about this area.

**Review Assessment: Checking Correctness Of Derivations And Theory:**

I assessed the sensibility of the derivations and theory.

**Review Assessment: Checking Correctness Of Experiments:**

I assessed the sensibility of the experiments.

**Review Assessment: Thoroughness In Paper Reading:**

I read the paper at least twice and used my best judgement in assessing the paper.

---

> ### Author Response · Authors · 2019-11-15
> **Response to Reviewer #4**
>
> Thank you for the comments and suggestions on the manuscript. Please find below our responses to your corresponding questions:
>
> 1. The training time for the proposed Reweighted RNN model (with our default settings, namely, the compressed sensing rate of 0.2, d=3 hidden layers and h= 2^10 hidden units per layer) is 3,521 seconds, which is higher than that of the l1-l1-RNN (Le et al. 2019); for the same settings, the training time of l1-l1-RNN is 2,985 seconds. We note however that comparing computational times in our experiments is not accurately indicating complexity as execution times depend heavily on the implementation of the model. Specifically, we used the Tensorflow implementations provided by the authors of the Independent-RNN, Fast-RNN and Spectral RNN models. The rest of the models were implemented in Pytorch; among these models, the vanilla RNN, LSTM, and GRU cells are written in CuDNN (default Pytorch implementations), so they are significantly faster in training (294s, 799s, and 655s, respectively, with the default settings) than the other models. It is, however, worth noting that our reweighted RNN model uses significantly fewer trainable parameters (4.47M in the default settings) compared to the popular variants of RNN (the vanilla RNN: 5.58M, stacked LSTM: 21.48M, and stacked GRU: 16.18M).
>
> 2. Thank you for this comment. Regarding the width of the network, the gain in performance is already quite minimal going from 2^11 to 2^12 neurons. Regarding the depth, we have trained a version of our network with 7 layers (with all other settings kept intact) and achieved a reconstruction performance of 38.5 dB in terms of PSNR. We can deduce that using a 6-layer version of the proposed Reweighted RNN model yields the best performance on Moving MNIST (with 8000 training samples).
>
> 3. We agree with the reviewer that experiments on other datasets except for Moving MNIST would be needed to demonstrate the full potential of the proposed network. Due to time limitations, we are unfortunately not able to report further extensive experiments in this paper. Nevertheless, in the Appendix, we have added further experimental evaluations of our model in popular RNN tests, namely, the pixel MNIST classification task, the adding task, and the copy task (see Appendix E).
>
> 4. At the moment, we are aware of the following limitations: (i) The extra trainable parameters present in Reweighted-RNN lead to an increase in the training time compared to the baseline l1-l1-RNN model (Le et al. 2019). (ii) Despite our goal of offering explainability in design, there are still several “black-box” components in our network, including the optimal choices for the number of layers and number of hidden units per layer. So far, these are still determined by experiments. (iii) Regarding the theoretical aspects of the paper, the derived generalization bounds - while being the first for deep RNN models - still depend on the network depth and width. In effect, the current bound (see Eq. (15)) is in the order of the square root of the network depth d multiplied by the number of time steps T, and also depends on the logarithm of the number of hidden units h. When increasing the number of parameters and/or the number of time steps T, the bound would be increased following the increase of depth/width/the number of time steps.

---

### Decision · Program_Chairs · 2019-12-19

**Decision:**

Reject

**Comment:**

This paper presents a novel RNN algorithm based on unfolding a reweighted L1-L1 minimization problem. Authors derive the generalization error bound which is tighter than existing methods.
All reviewers appreciate the theoretical contributions of the paper, particularly the derivation of generalization error bounds. However, at a higher-level, the overall idea is incremental because RNN by unfolding L1-L1 minimization problem (Le+,2019) and reweighted L1 minimization (Candes+,2008) are both known techniques. The proposed method is essentially a simple combination of them and therefore the result seems somewhat obvious. Also, I agree with reviewers that some experiments are not deep enough to support the theory. For example, for over-parameterization (large model parameters) issue, one can compare the models with the same number of parameters and observe how they generalize.
Overall, this is the very borderline paper that provides a good theoretical contribution with limited conceptual novelty and empirical evidences. As a conclusion, I decided to recommend rejection but could be accepted if there is a room.